# Specialized contributions of mid-tier stages of dorsal and ventral pathways to stereoscopic processing in macaque

Toshihide W Yoshioka[1,2†], Takahiro Doi[3]*, Mohammad Abdolrahmani[4], Ichiro Fujita[1,2]*

[1]Laboratory for Cognitive Neuroscience, Graduate School of Frontier Biosciences, Osaka University, SuitaOsaka, Japan; [2]Center for Information and Neural Networks, Osaka University and National Institute of Information and Communications Technology, SuitaOsaka, Japan; [3]Department of Psychology, University of Pennsylvania, Philadelphia, United States; [4]Laboratory for Neural Circuits and Behavior, RIKEN Center for Brain Science (CBS), Wako, Japan

**Abstract** The division of labor between the dorsal and ventral visual pathways has been well studied, but not often with direct comparison at the single-neuron resolution with matched stimuli. Here we directly compared how single neurons in MT and V4, mid-tier areas of the two pathways, process binocular disparity, a powerful cue for 3D perception and actions. We found that MT neurons transmitted disparity signals more quickly and robustly, whereas V4 or its upstream neurons transformed the signals into sophisticated representations more prominently. Therefore, signaling speed and robustness were traded for transformation between the dorsal and ventral pathways. The key factor in this tradeoff was disparity-tuning shape: V4 neurons had more even-symmetric tuning than MT neurons. Moreover, the tuning symmetry predicted the degree of signal transformation across neurons similarly within each area, implying a general role of tuning symmetry in the stereoscopic processing by the two pathways.

**\*For correspondence:**
doi.takah@gmail.com (TD);
fujita@fbs.osaka-u.ac.jp (IF)

**Present address:** [†] Brain Science Institute, Tamagawa University, Tokyo, Japan

**Competing interests:** The authors declare that no competing interests exist.

## Introduction

The dorsal and ventral pathways of the primate visual system have served as a widely influential model of parallel sensory processing in the mammalian cerebral cortex (*Ungerleider and Mishkin, 1982*). Similar organizational structures have been found in the visual cortical networks of cats and mice (*Hilgetag et al., 2000*; *Wang et al., 2012*). Moreover, related ideas have been put forth regarding non-visual domains such as auditory (*Rauschecker and Scott, 2009*), somatosensory (*Dijkerman and de Haan, 2007*), and language-related processes (*Hickok and Poeppel, 2007*; *Saur et al., 2008*). Despite their widespread influence, direct comparison of the dorsal and ventral pathways at the single-neuron resolution with matched stimuli is surprisingly scarce (*Cheng et al., 1994*; *Lehky and Sereno, 2007*), leaving open the exact nature of the differences and similarities between the two pathways.

In the visual systems of human and non-human primates, the dorsal pathway is implicated in spatial vision and vision for action, whereas the ventral pathway is implicated in object recognition and vision for perception (*Goodale and Milner, 1992*; *Ungerleider and Mishkin, 1982*). Contrary to initial speculation, both pathways process the same visual cues. Neurons with shape and color selectivity are found not only in ventral areas but also in dorsal areas (*Seidemann et al., 1999*; *Sereno and Maunsell, 1998*). Likewise, neurons with motion direction selectivity are found not only in dorsal areas but also in ventral areas (*Desimone and Schein, 1987*; *Li et al., 2013*; *Mountcastle et al.,*

*1987*). Investigating how the dorsal and ventral pathways process the same cue at the single-neuron resolution is critical to better understand the parallel processing strategies in the visual system.

One of the most notable visual cues in this regard is binocular disparity, the positional difference between images projected to the left and right eyes. As a precise cue for stereoscopic depth, binocular disparity is used for many functions, including the recognition of 3D objects, visually guided 3D action (e.g., reaching, grasping, and vergence eye movement), and navigation through the 3D environment (e.g., moving through obstacles). As binocular disparity is used for both 'what' (object vision) and 'where' (spatial vision) functions, the visual system processes binocular disparity along both the dorsal and ventral pathways (*Neri, 2005*; *Orban et al., 2006*; *Parker, 2007*; *Roe et al., 2012*; *Welchman, 2016*). A few studies directly compared the disparity processing of the two pathways with matched visual stimuli. These studies examined the choice-related activity of multiple neurons (*Verhoef et al., 2010*) or the behavioral effects of electrical microstimulation (*Verhoef et al., 2012*; *Verhoef et al., 2015*). In the current study, we focused on disparity processing at the single-neuron resolution.

To encode binocular disparity faithfully in accordance with the geometry of the 3D world, the visual system should identify the corresponding visual features in left and right retinal images that originate from the same surface point in the 3D space (*Julesz, 1971*; *Marr and Poggio, 1976*; *Marr and Poggio, 1979*). A conventional method to probe the neuronal process for stereo correspondence is to measure the disparity selectivity for anticorrelated random-dot stereograms (aRDSs) (*Chen et al., 2017*; *Cumming and Parker, 1997*; *Goncalves and Welchman, 2017*; *Janssen et al., 2003*; *Krug et al., 2004*; *Kumano et al., 2008*; *Nieder and Wagner, 2001*; *Ohzawa, 1998*; *Samonds et al., 2013*; *Takemura et al., 2001*; *Tanabe et al., 2004*; *Theys et al., 2012*). In aRDSs, one eye image is contrast-reversed to eliminate the natural solution to the correspondence problem; the natural solution requires the depth of the corresponding points to vary smoothly over space, but such a solution does not exist for aRDSs. Accordingly, the neural correlate of the correspondence solution should have no or minimal disparity selectivity for aRDSs. Although the activity of many V1 neurons encodes binocular disparity, these neurons also falsely encode the disparity embedded in aRDSs (*Cumming and Parker, 1997*). Simple computations akin to interocular cross-correlation explain this primitive disparity representation (*Ohzawa et al., 1990*). To solve the correspondence problem, a more complex computation should ensue and transform the primitive correlation-based disparity representation into the sophisticated representation that is based on binocularly matching features ('match-based representation').

In this study, we directly compared the disparity representations in areas MT and V4 of the monkey. MT and V4 are counterpart mid-tier stages of the dorsal and ventral pathways, both of which causally contribute to stereoscopic depth judgment (*DeAngelis et al., 1998*; *Shiozaki et al., 2012*; *Uka and DeAngelis, 2006*). We used a recently developed extension of aRDSs, called graded anticorrelation (*Doi and Fujita, 2014*; *Doi et al., 2011*; *Doi et al., 2013*). Moreover, we used virtually identical experimental and analysis methods (and one identical animal) in MT and V4 experiments to achieve a more controlled comparison of the two areas than has previously been attempted. We found that the disparity selectivity was indistinguishable between MT and V4 in their responses to the conventional stimuli, aRDSs. However, graded anticorrelation revealed that the disparity representation in V4 was more strongly specialized for the correspondence solution than that in MT. The relative advantage of MT's disparity signaling was its speed and strength. These results shed light on the division of labor between the dorsal and ventral pathways. The dorsal pathway carries out rapid signaling to produce timely behavioral outputs, whereas the ventral pathway emphasizes complex computations to derive sophisticated sensory representations. We also found that the disparity-tuning shape was highly consistent with the degree of neuronal specialization: neurons with more even-symmetric tuning had responses closer to the correspondence solution both within and across MT and V4. We will discuss this relationship in light of the adaptation of the stereoscopic system to natural binocular inputs (*Haefner and Cumming, 2008*).

# Results

## Dissociating correlation-based and match-based representations by graded anticorrelation

We analyzed the neuronal responses to graded anticorrelation to characterize how well these responses conformed to the solution of the correspondence problem; that is, how much the disparity tuning of a visual neuron deviated from the simple correlation-based representation toward a more sophisticated match-based representation (*Doi and Fujita, 2014*; *Doi et al., 2011*; *Doi et al., 2013*). For this assessment, we manipulated the binocular correlation of RDSs by reversing the luminance contrast of a varying proportion of dots in one eye (*Figure 1A*). This stimulus manipulation systematically changed the binocular correlation (correlation between the left-eye and right-eye images; the top arrow in *Figure 1A*) and the binocular match (the percentage of binocularly matched features; the bottom arrow in *Figure 1A*) in a dissociable manner. At one end, correlated RDSs (cRDSs) had 100% correlation and 100% match (*Figure 1A* right). However, at the other end, aRDSs had −100% correlation and 0% match, indicating complete dissociation (*Figure 1A* left). In the middle, half-matched RDSs (hmRDSs) showed that half of the dots were contrast-reversed in one eye. Thus, the overall binocular correlation of hmRDSs was 0% since correlated and anticorrelated dots canceled out, while the strength (proportion) of the binocular match was 50%, indicating dissociation in the opposite direction.

We used these dissociated values of binocular correlation and binocular match to derive the predictions of disparity-tuning curves. A disparity detector with the pure correlation-based representation should lose its disparity selectivity and show flat tuning for hmRDS, because hmRDSs offer no binocular correlation that is necessary for these kinds of detectors to encode disparity (*Figure 1B* left, green line). For aRDSs, the same detectors should have an inverted tuning curve of the same magnitude as the curve for cRDSs, because of the sign inversion of the stimulus correlation (*Figure 1B* left, pink curve). By contrast, ideal match-based detectors should retain disparity tuning with a decreased amplitude for hmRDSs but should lose disparity selectivity for aRDSs (*Figure 1B* right). This is because hmRDSs contain binocularly matched features whereas aRDSs offer no correctly matching features between the two eyes (i.e., no natural solution to the correspondence problem). The disparity energy model is a classic implementation of the correlation-based detector (*Cumming and Parker, 1997*; *Ohzawa et al., 1990*). Additional nonlinearity can transform the correlation-based detector to the match-based detector under some conditions (*Doi and Fujita, 2014*; *Henriksen et al., 2016a*).

## Disparity selectivity in MT and V4 is biased toward correlation-based and match-based representations, respectively

We recorded single-neuron activity from monkeys while they performed a fixation task. We compared how neurons in MT and V4 responded to graded anticorrelation by examining the

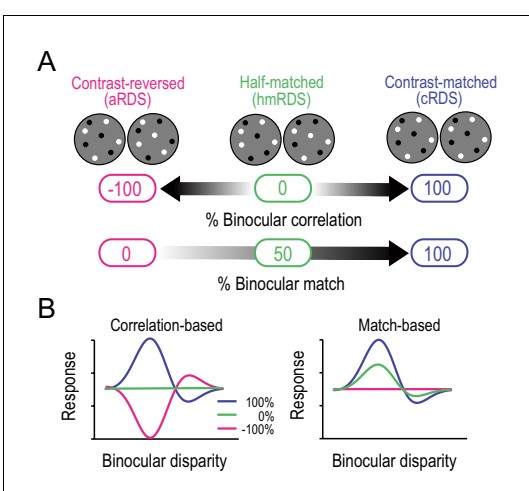

**Figure 1.** Correlation-based and match-based representations predict distinct responses to graded anticorrelation. (**A**) Three RDSs with graded anticorrelation. From right to left, the fraction of binocularly contrast-matched dots decreases from 100% (correlated RDS or cRDS) through 50% (half-matched RDS or hmRDS) to 0% (anticorrelated RDS or aRDS). This same stimulus manipulation decreases binocular correlation gradually from 100% (cRDS) through 0% (hmRDS) to −100% (aRDS). (**B**) Hypothetical disparity-tuning curves to graded-anticorrelation stimuli as predicted by correlation-based or match-based disparity representations. In the correlation-based responses (left), the absolute value and the sign of the percent binocular correlation determine the amplitude and shape of the tuning function, respectively. In the match-based responses (right), the amplitude reflects the fraction of matched dots, and the shape is invariant.

responses of 83 disparity-selective neurons in MT (monkey O, N = 32; monkey A, N = 51) and 78 disparity selective neurons in V4 (monkey O, N = 29; monkey I, N = 49). Disparity selectivity was determined from the responses to cRDSs (Kruskal–Wallis test; p<0.05). Monkey O provided data sets for both MT and V4. The sample size of disparity-selective neurons was determined based on a similar study conducted in V4 (*Tanabe et al., 2004*).

The graded anticorrelation of RDSs differentially modulated the disparity tuning of neurons in MT and V4. *Figure 2* shows an example of single neurons demonstrating characteristic differences between the MT and V4 populations. Both of the example neurons responded maximally at 'near' disparities with 100% binocular correlation, but the decrease in the correlation level changed the tuning curves differentially between them. The change observed for the example MT neuron is reminiscent of the correlation-based representation of disparity: when the correlation level decreased from 100% to 0% (hmRDS), the tuning amplitude gradually decreased to almost zero (flat curve). A further decrease in the correlation level from 0% to −100% (aRDSs) recovered the tuning but with an inverted shape (*Figure 2A*). Compared to this MT example, the response pattern of the example V4 neuron was closer to the match-based representation: the tuning amplitude was noticeably retained at 0% correlation with a relatively unchanged tuning phase (shape). The tuning was more attenuated than the MT example at −100% correlation, without the exact shape inversion (*Figure 2B*). We plotted these responses in the 2D plane defined by binocular disparity and correlation level. The same pattern was also apparent when we compared the observed 2D response profiles (*Figure 2C,D*) to

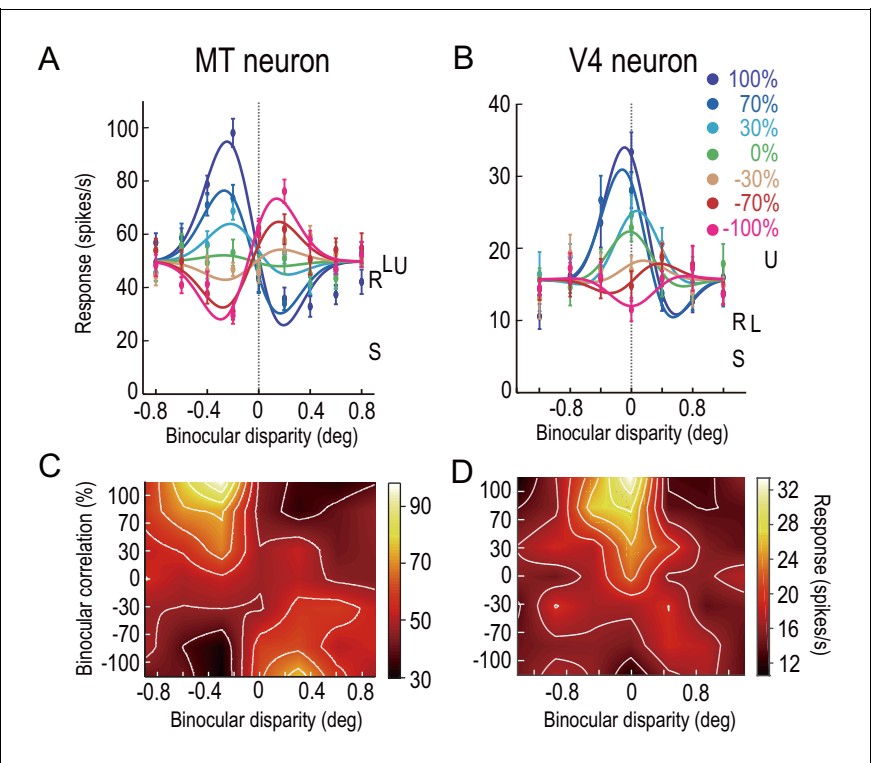

**Figure 2.** Disparity-tuning curves of example neurons in MT and V4. The average firing rates of example neurons at different binocular-disparity values and binocular-correlation levels in MT (**A**) and V4 (**B**). Different colors indicate different correlation levels. Error bars indicate the ± SEMs. Letters S, U, R, and L indicate the level of spontaneous (pre-stimulus) activity, the response to uncorrelated RDSs, and the responses to right and left monocular images, respectively. (**C** and **D**) Color plots of responses of the neurons shown in **A** and **B** in a 2D plane defined by binocular disparity and correlation level. Brighter colors indicate stronger responses, as shown in the scale bars. The raw responses are linearly interpolated in both dimensions. The V4 example was previously shown in *Abdolrahmani et al., 2016*.

The online version of this article includes the following source data and figure supplement(s) for figure 2:

**Source data 1.** Tuning-curve data of example neurons.
**Figure supplement 1.** Predicted response maps for example MT and V4 neurons.

the predicted profiles from the correlation-based and match-based representations (*Figure 2—figure supplement 1*). Below we quantify and validate this impression with both model-free and model-based analyses.

The disparity representation was drastically different between neurons in MT and V4 at the population level. MT neurons retained the characteristics of the correlation-based disparity representation presumably constructed in V1. These features were largely eliminated in V4, where the disparity selectivity was strongly biased toward the match-based representation. *Figure 3* shows the results of model-free population analyses without curve fitting. First, we analyzed the fraction of MT neurons with significant disparity tuning at each correlation level (Kruskal–Wallis test, p<0.05 tested independently at each correlation level below 100%). The fraction had a characteristic U shape consistent with the correlation-based representation: disparity encoding relies on the nonzero binocular correlation of the stereo images (*Figure 3A*). The fraction was minimal at −30% correlation, where only one out of 83 MT neurons (1.2%) was disparity selective. With further decrease from −30% to −100%, the fraction increased to 21/83 (25.3%). The pure correlation-based model posits that the

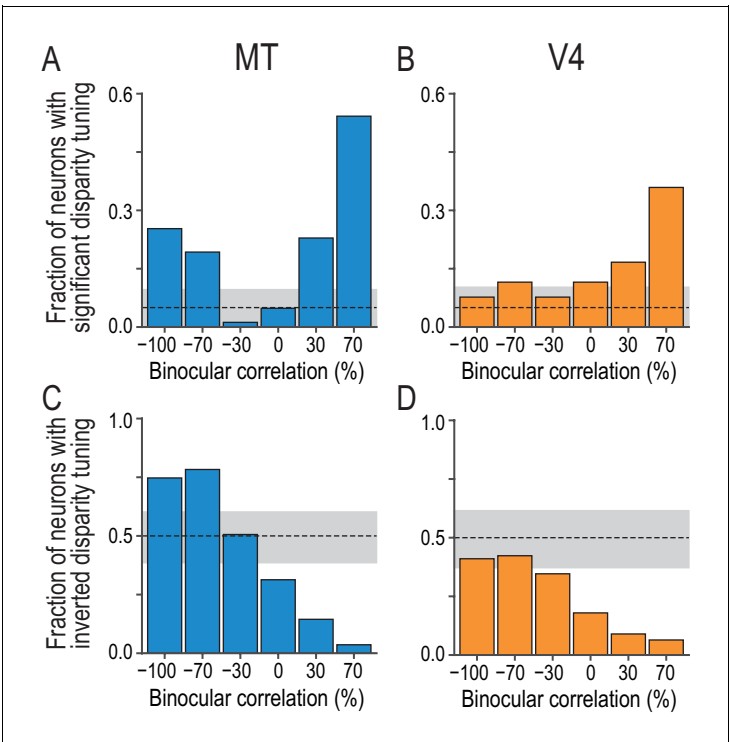

**Figure 3.** Fractions of neurons with significant and inverted disparity tuning show different dependencies on graded anticorrelation between MT and V4. (**A and B**) The fraction of neurons with significant disparity tuning as a function of binocular correlation (Kruskal–Wallis test; p<0.05). The population comprises all recoded neurons with significant tuning at 100% correlation (83 and 78 neurons for MT and V4, respectively). The dashed line indicates the chance level and the gray area indicates its 95% interval based on the binomial distribution. The fraction at 100% correlation is one by definition. (**C and D**) The fraction of neurons with inverted disparity tuning (relative to the tuning to cRDSs) as a function of binocular correlation. Negative tuning correlation was considered to indicate tuning inversion. The same populations as in (**A and B**) were analyzed. (See *Figure 3—figure supplement 1* for the same analysis applied to the subpopulations of tuning curves without significant selectivity; we suggest that the supplementary results warrant the inclusion of non-selective tuning in this main analysis.) The chance level is 0.5. The fraction at 100% correlation is zero by definition.

The online version of this article includes the following source data and figure supplement(s) for figure 3:

**Source data 1.** The p-values for Kruskal–Wallis test to assess disparity selectivity and tuning-curve correlation to assess tuning inversion at each binocular-correlation level.

**Figure supplement 1.** Fractions of neurons with tuning inversion in a subpopulation of neurons with nonsignificant disparity tuning.

fraction should be one at −100% correlation and the minimum at 0% correlation. The mismatch slightly hints at the match-based representation, although overall the disparity selectivity in MT is correlation-based.

The tuning shape of MT neurons also exhibited the characteristic of the correlation-based representation: negative binocular correlation inverts the tuning shape. We detected tuning inversion based on a negative correlation coefficient between the responses to cRDSs and those at a lower binocular correlation level. In MT, the fraction of the neurons with inverted tuning gradually increased as binocular correlation decreased (*Figure 3C*). Notably, significant fractions of neurons inverted their tuning curves at the two most negative correlation levels we tested (75% [62/83] and 78% [65/83] of neurons at the −100% and −70% correlation levels, respectively; p=3.8 × 10$^{-6}$ and 1.1 × 10$^{-7}$; H$_0$: the fraction is 0.5; binomial test). We also observed some deviation from the pure correlation representation: the fraction of the tuning inversion was not one at the −100% correlation level. The fraction was closest to 0.5 at the −30% correlation level, echoing the finding that the fraction of disparity-selective neurons was minimal at −30% (*Figure 3A*).

The V4 population data were markedly different from the MT data, entirely lacking the aforementioned characteristics of the correlation-based representation. Instead, the disparity selectivity of V4 neurons was mostly consistent with the match-based representation. The fraction of neurons with significant disparity tuning did not demonstrate the characteristic U shape of the correlation-based representation, but gradually decreased with gradual anticorrelation (*Figure 3B*), mirroring the decrease in the binocular-match level (*Figure 1A*). The fraction of neurons with inverted tuning gradually increased with gradual anticorrelation but did not surpass the chance level (*Figure 3D*). Just as the MT data were not completely consistent with the pure correlation-based representation, the V4 data were not completely consistent with the pure match-based representation. Aggregating the population responses in V4 would help create the pure match-based representation in downstream areas (see Discussion).

Next, we characterized how each individual neuron in MT and V4 represented binocular disparity. To this end, we extended a frequently used metric, the amplitude ratio, which quantifies the disparity-tuning amplitude at −100% correlation (aRDS) relative to that at 100% correlation (cRDS; *Cumming and Parker, 1997*; *Krug et al., 2004*; *Kumano et al., 2008*; *Samonds et al., 2013*; *Tanabe et al., 2004*). First, we added a sign to the amplitude ratio, where a negative sign indicates tuning inversion. Second, we calculated the signed amplitude ratio for all intermediate correlation levels, not just for −100%. This signed amplitude-ratio function should have different shapes for the correlation-based and match-based neurons (*Figure 4A*). For the correlation-based neurons, it should decrease from 1 through 0 to −1 as binocular correlation decreases from 100% through 0% to −100%. For the match-based neurons, it should decrease as the correlation level decreases but should always stay above 0 except at −100% correlation. We do not argue that the match-based prediction must be exactly linear as shown in *Figure 4A*; the linear decrease is just the simplest example of monotonic functions.

We devised a new metric called the area ratio that takes the value of 1 for the correlation-based neurons and 0 for the match-based neurons (*Figure 4B*). The numerator of the area ratio is the area (integral) of the signed amplitude-ratio function across the range where the function is negative (inverted tuning); the denominator is the area of the range where the function is positive. Therefore, the area ratio quantifies how strongly each neuron encodes the disparity with the inverted tuning shape as opposed to the non-inverted tuning shape. The example MT and V4 neurons shown in *Figure 2* had the area ratios of 0.79 and 0.15, respectively (*Figure 4C,D*).

The distributions of the area ratio markedly differed between V4 and MT (density distributions, *Figure 5A,B*; cumulative distributions, *Figure 5C*). The distribution for V4 was strongly skewed toward zero, whereas the distribution for MT was less skewed. The V4 distribution was unimodal, whereas the MT distribution might have been bimodal, with an approximately normal distribution centered at around 0.6 and another potentially distinct peak at around 0. Although the MT distribution might actually have been unimodal (Hartigan's dip test for bimodality: dip index = 0.053, p=0.24, N = 10,000), the bootstrap distribution for the peak location consisted of two distinct clusters for MT (*Figure 5—figure supplement 1A*). Quantile–quantile plots revealed that the V4 and MT data were close to exponential and normal distributions, respectively, within the interquartile range (*Figure 5—figure supplement 2*). The shape difference was statistically significant (*Figure 5C*; Kolmogorov–Smirnov test, p=7.9 × 10$^{-5}$). In MT, the median area ratio was 0.50 and significantly larger

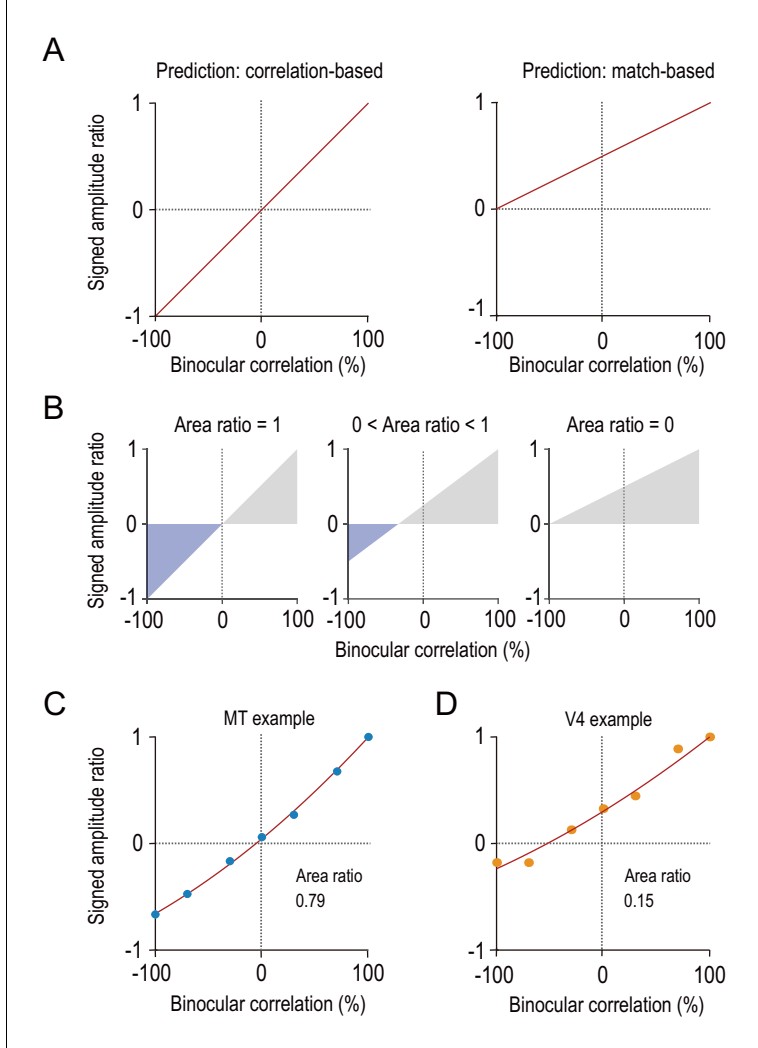

**Figure 4.** Characterizing disparity representation for each individual neuron. (**A**) Correlation-based and match-based disparity representations predict different profiles for the signed amplitude ratio as a function of binocular correlation. The signed amplitude ratio at each correlation level reflects both how much the tuning amplitude is reduced and if the tuning shape is inverted by partial anticorrelation. (**B**) Area ratio as a novel metric to quantify how well the observed responses conform to correlation-based or match-based prediction. The area ratio was defined as the ratio of two areas: the numerator was the area with the negatively signed amplitude ratios (blue); the denominator was the area with the positively signed amplitude ratios (gray). An area ratio of 1 indicates perfectly correlation-based responses (*leftmost*), and an area ratio of 0 indicates perfectly match-based responses (*rightmost*). An intermediate area ratio indicates an intermediate representation type (*middle*). (**C and D**) Area ratios calculated for the example MT and V4 neurons shown in *Figure 2*. The signed amplitude ratios were calculated based on the best fitted Gabor tuning functions (see Materials and methods). Then, quadratic functions were fitted to interpolate the data points.

The online version of this article includes the following source data for figure 4:

**Source data 1.** Signed amplitude ratio of example neurons.

than 0 (two-sided Wilcoxson signed-rank test, p=1.6 × $10^{-10}$). In V4, it was 0.17 and significantly lower than that in MT (two-sided Mann–Whitney *U*-test, p=3.6 × $10^{-4}$).

We confirmed the significant difference in the median area ratio between MT and V4 in three control analyses. First, we calculated a model-free version of area ratio without using any fitting (*Figure 5—figure supplement 3A*). Second, we used the data obtained only from the same individual monkey (*Figure 5—figure supplement 3B*). Third, we removed the neurons with particular types of non-monotonic, signed-amplitude ratio functions that could complicate the interpretation of area

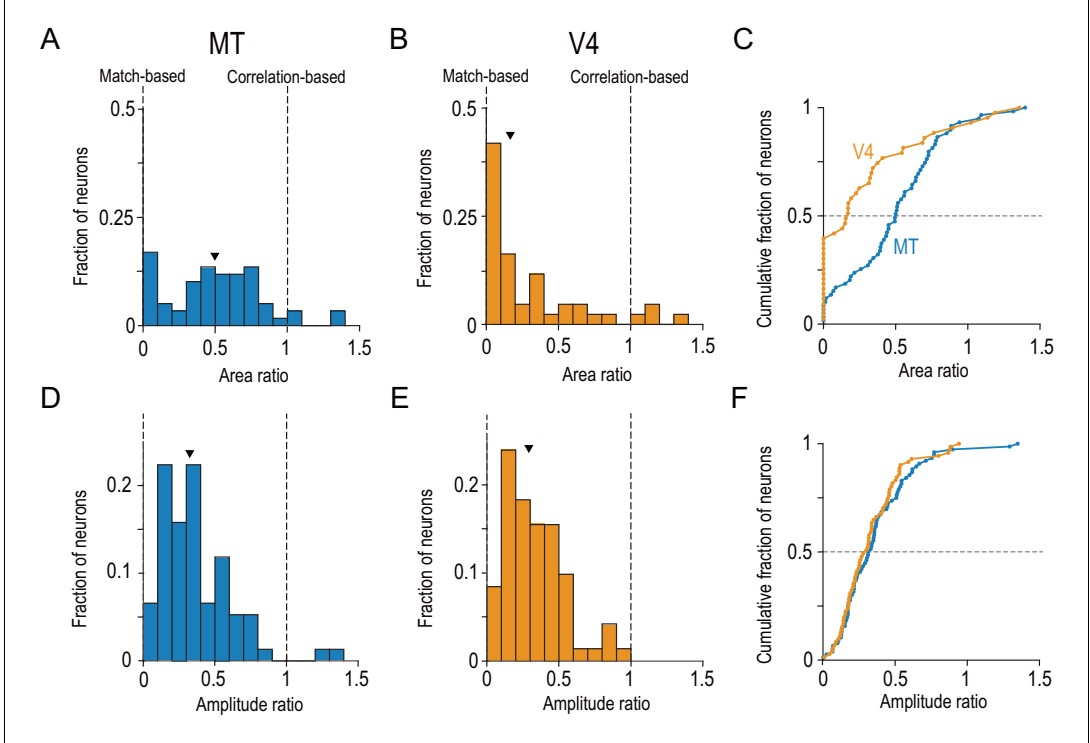

**Figure 5.** Distinct distributions of single-neuron disparity representation between MT and V4 revealed only by area ratio. Distributions of the area ratio, our new metric (A–C), and the amplitude ratio, the conventional metric (D–F), are shown. A–C consist of the neurons with good Gabor-function fitting to disparity-tuning curves and good quadratic-function fitting to signed amplitude ratios ($R^2$ >0.6; MT, N = 59; V4, N = 43). D–F consist of the neurons with good Gabor-function fitting ($R^2$ >0.6; MT, N = 76; V4, N = 71). Triangles indicate the medians. C and F show cumulative distributions of the area ratio and amplitude ratio, respectively. The amplitude ratio histogram for V4 (E) is a replot of the result reported in *Abdolrahmani et al., 2016*.

The online version of this article includes the following source data and figure supplement(s) for figure 5:

**Source data 1.** Data for *Figure 5* and its figure supplements.
**Figure supplement 1.** Bootstrap distribution for peak location of area-ratio distribution.
**Figure supplement 1—source data 1.** Bootstrap distribution of the area ratio's peak-bin locations.
**Figure supplement 2.** The distributions of the area ratios for MT and V4 are compared against normal and exponential distributions, respectively.
**Figure supplement 3.** The differential distributions of the area ratios between V4 and MT hold in two control analyses.
**Figure supplement 4.** Neurons with V-shape signed amplitude-ratio functions.
**Figure supplement 4—source data 1.** Signed amplitude ratios from Gabor-function disparity tuning.
**Figure supplement 5.** The area ratio is correlated neither with the strength of disparity selectivity nor with the responsiveness in MT and V4.

ratio (*Figure 5—figure supplement 4*, Appendix 1—Figure 5—text supplement 1). These results suggest that MT and V4 are qualitatively different in terms of the composition of disparity-selective neurons. In area V4, many single neurons are strongly specialized for the match-based representation. In area MT, the distribution is wider: some have the match-based representation but many have representations that are intermediate between fully match-based and fully correlation-based.

Critically, the difference between MT and V4 was revealed only by the area ratio and not by the amplitude ratio, the latter of which is the conventional metric based only on the responses to cRDSs and aRDSs. The distribution of the amplitude ratio was indistinguishable between the two areas (*Figure 5D–F*). The median values were 0.32 and 0.30 for MT and V4, respectively (two-sided Mann–Whitney $U$-test, p=0.47). The results of this control analysis are instructive, because although there were previous speculations regarding the difference between MT and V4 in disparity representation (*Parker, 2007*), these were based only on the neuronal responses to cRDSs and aRDSs (*Krug et al., 2004*; *Tanabe et al., 2004*). These previous data sets hint at more strongly reduced disparity selectivity for aRDSs in V4 than in MT (compare Figure 3C in *Krug et al., 2004* and Figure 6A in *Tanabe et al., 2004*), but this apparent difference between MT and V4 might arise from differences in the cell-selection procedures used: the MT study excluded the cells without the significant

disparity selectivity for aRDSs from the figure, whereas the V4 study did not. Applying the same statistical methods to MT and V4, we found that full anticorrelation reduced disparity selectivity similarly for MT and V4.

The use of non-moving RDSs was unlikely to bias our estimation of the area ratios in MT, even though these RDSs might not drive MT neurons to their maximum responsiveness. One might argue that the use of non-coherent-motion RDSs in our experiments resulted in poorer disparity selectivity and led to a biased estimation toward the correlation-based representation in MT. This explanation is possible because stimuli with non-preferred motion parameters decrease the strength of disparity selectivity for cRDSs in MT (*Palanca and DeAngelis, 2003*). At least in V1, neurons with weaker disparity selectivity for cRDSs exhibit responses closer to the correlation-based prediction (*Henriksen et al., 2016b*). However, we found no correlation between area ratio and the disparity selectivity for cRDSs (MT: Spearman's correlation coefficient, $r_s = -0.25$, p>0.05; V4: $r_s = -0.25$, p>0.1; *Figure 5—figure supplement 4A,B*), or between area ratio and the baseline response (MT: $r_s = -0.12$, p>0.3; V4: $r_s = -0.08$, p>0.6; *Figure 5—figure supplement 4C,D*). These results suggest that neither the strength of the disparity selectivity nor the responsiveness is strongly related to which type of disparity representation a neuron exhibits in MT and V4.

## Disparity-tuning symmetry predicts disparity representation in MT and V4

Can we predict if a disparity-selective neuron has a match-based representation or correlation-based representation based on its responses to cRDSs? If so, what response feature carries the information about the representation type? We reported above that the strength of disparity selectivity or responsiveness was not correlated with the area ratio. Here, we examined the shape of the disparity-tuning curve obtained with cRDSs. The tuning-curve symmetry has been of particular interest because it reflects the underlying mechanism of disparity detection and implies the utility of the detected signals (*Cumming and DeAngelis, 2001*; *DeAngelis et al., 1991*; *Marr and Poggio, 1979*; *Poggio and Fischer, 1977*; *Read and Cumming, 2007*; *Goncalves and Welchman, 2017*).

We found a strikingly general relationship between the symmetry of the disparity-tuning curve and the type of disparity representation: neurons with even-symmetric tuning tended to have the match-based representation within both MT and V4. To quantify the symmetry of a tuning curve, we devised a metric that monotonically encodes the tuning-curve symmetry (see reflected symmetry phase in the Materials and methods section; see *Read and Cumming, 2004*) for symmetry phase. A larger value of the reflected symmetry phase indicates a stronger odd symmetry, with 0 for the pure even-symmetric shape ('tuned excitatory' or 'tuned inhibitory' types by a conventional classification) and $\pi/2$ for the pure odd-symmetric shape ('near' or 'far' types). The area ratio was positively correlated with the reflected symmetry phase both in MT and V4 (*Figure 6A,B*; MT: $r_s = 0.41$, p=$1.4 \times 10^{-3}$; V4: $r_s = 0.34$, p=0.025). The analyses based on the fitted Gabor phase instead of the reflected symmetry phase confirmed the same conclusion (*Figure 6—figure supplement 1*). Note that the observed correlation was only moderate for both MT and V4. The best-fitted linear model accounted for 24% of the total variance in the area ratio for MT and 20% for V4 (*Figure 6—figure supplement 2A*, top row). These percentages increased only to 25% and 31% with the quadratic model in MT and V4, respectively (*Figure 6—figure supplement 2A*, bottom row). These results imply that the tuning symmetry is only part of the explanation for the cell-by-cell variation in disparity representations. Consistent with this view, the width of the spatial-frequency tuning is correlated with the amplitude ratio in V4 (*Kumano et al., 2008*).

The same relationship also dictated the overall difference between neurons in MT and V4. We showed that neurons in MT had larger area ratios than those in V4 (*Figure 5C*). We found that neurons in MT had more strongly odd-symmetric tuning curves than those in V4 by comparing their distributions of the reflected symmetry phase (*Figure 7*). In both MT and V4, the phase was biased toward even-symmetric shapes: the histograms had peaks near 0 (*Figure 7A,B*). However, the distribution for MT was less strongly biased toward the even shape (a reflected symmetry phase of 0) than that for V4. The difference was clearer in the cumulative distribution (*Figure 7C*). Although the median values happened to be almost identical ($0.074\pi$ and $0.073\pi$ for MT and V4, respectively), the cumulative distribution for V4 was consistently biased toward smaller values than that for MT. Furthermore, the quantiles of these two distributions had a linear relationship with a slope value of 1.39 (*Figure 7D*; SE = 0.015; $R^2 = 0.98$; MT quantiles plotted against V4 quantiles), meaning that

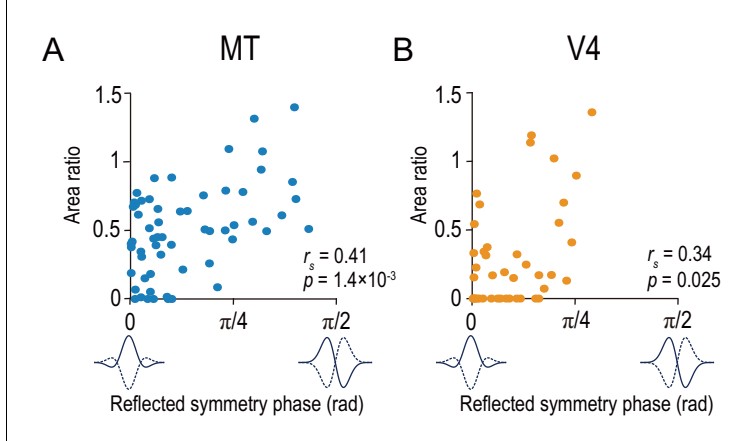

**Figure 6.** Area ratio is correlated with tuning-curve symmetry in both MT and V4. Area ratios of neurons in MT (**A**) and V4 (**B**) were plotted against the reflected symmetry phase, which takes the value of 0 and $\pi/2$ for even-symmetric and odd-symmetric tuning curves, respectively. The symmetry was computed for the responses to correlated RDSs. The plots consist of the neurons with good fitting quality ($R^2 > 0.6$ both for Gabor functions for disparity-tuning curves and quadratic functions for the signed amplitude ratio; N = 59 and 43 for MT and V4, respectively). $r_s$ indicates Spearman's rank correlation coefficient.

The online version of this article includes the following source data and figure supplement(s) for figure 6:

**Source data 1.** Data for *Figure 6* and its figure supplement.

**Figure supplement 1.** The area ratio is correlated with the symmetry of the disparity-tuning curve as quantified by the Gabor function's phase parameter in MT and V4.

**Figure supplement 2.** Linear and quadratic regressions for area ratio as dependent variable and reflected symmetry phase as independent variable.

**Figure supplement 2—source data 1.** Best-fitted regression coefficients.

multiplying the reflected symmetry phases for the V4 data set by 1.39 brings the V4 and MT distributions into alignment. Therefore, the disparity tunings in V4 had more even-symmetric shapes than those in MT. At the same time, the disparity representations in V4 were more match-based than those in MT. These results yield a strikingly general principle of the stereoscopic system both within and across the dorsal area MT and ventral area V4: neurons with even-symmetric disparity-tuning curves preferentially constitute the neural solution to the correspondence problem compared to those with odd-tuning curves.

## MT exhibits stronger and faster disparity selectivity than V4 for correlated RDSs

What is the advantage of the disparity signals transmitted by MT? We showed that MT had a more primitive, correlation-based disparity representation than V4, which manifested in how the disparity selectivity changed with graded anticorrelation. Here we show that the disparity selectivity for cRDSs is faster and stronger in MT than in V4 (*Figure 8*). We devised a metric called signed disparity discrimination index (sDDI) and computed it using a sliding time window both before and during the stimulus presentation (10 ms width and step; see *Prince et al., 2002* for DDI). The absolute value of sDDI indicates the strength of the selectivity relative to the response noise: 0 for no selectivity and 1 for extremely large selectivity compared to the noise level. The positive and negative signs indicate whether the instantaneous selectivity quantified over a short time window is consistent with or opposite from the overall selectivity quantified over the whole stimulus duration, respectively.

We found that the average sDDI of MT neurons rose faster and reached a higher level than that of V4 neurons (*Figure 8A*). We fitted the cumulative Gaussian function to estimate the latency and amplitude of the disparity selectivity (mean and amplitude parameters, respectively; *Figure 8B*). The latency was 78 ms in MT and 101 ms in V4 (*Figure 8C*). The peak sDDI was 0.18 in MT and 0.13 in V4. Thus, compared with MT, in V4 the latency was 29% greater and the strength was 28% lower. The differences were statistically significant (p=0.0083 and p<1.0 $\times$ 10$^{-4}$ for the latency and

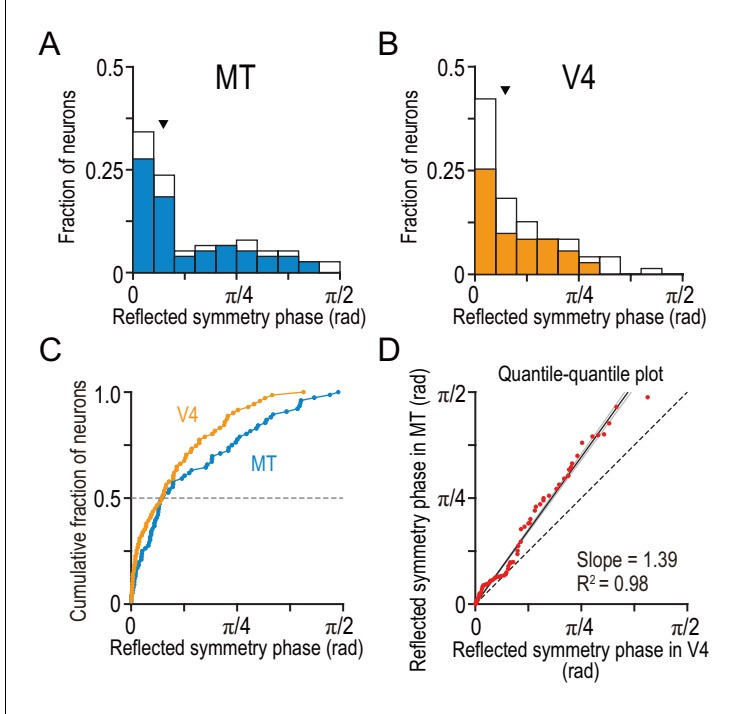

**Figure 7.** Tuning-curve symmetry is more strongly biased toward the even shape in V4 than in MT. (**A and B**) The distributions of the reflected symmetry phase. We included the neurons for which Gabor functions were well fitted to the disparity tuning ($R^2$ >0.6; N = 76 and 71 for MT and V4, respectively). The parts of the histograms with solid color correspond to the neurons for which quadratic functions were well fitted to the signed amplitude ratio (i.e., the same subpopulations as plotted in *Figure 6* and *Figure 5A–C*). Triangles indicate the medians. (**C**) Cumulative distributions of the reflected symmetry phase for the full populations shown in A and B. (**D**) Quantile–quantile plot comparing the shape of the distribution between V4 and MT. N = 71 (the number of V4 neurons, which was smaller than that of MT neurons). The horizontal axis plots the sorted values of the reflected symmetry phase for V4. The vertical axis plots the interpolated values of the reflected symmetry phase for MT at the corresponding quantiles. The black line indicates the linear regression. The flanking gray lines indicate the 95% confidence interval of the linear regression.

The online version of this article includes the following source data for figure 7:

**Source data 1.** The source file contains reflected symmetry phase for a larger population of neurons than analyzed in *Figure 6*.

---

amplitude differences between MT and V4, respectively; 10,000 bootstrap resamples). These results were surprising given that we used the RDSs suitable for driving V4 neurons (*Abdolrahmani et al., 2016*; *Kumano et al., 2008*; *Shiozaki et al., 2012*; *Tanabe et al., 2004*; *Tanabe et al., 2005*; *Umeda et al., 2007*). For example, unlike previous MT studies (*DeAngelis and Uka, 2003*; *Krug et al., 2004*), our RDSs did not contain visual motion, and their dot patterns were updated only slowly (10.6 Hz). The fast and robust disparity signals in MT should be useful for rapid three-dimensional eye movements (*Masson et al., 1997*) and urgent depth judgments in response to correlated stimuli.

## Disparity representation of V4 neurons near the fovea

The ventral pathway plays important roles in foveal vision (*Sheth and Young, 2016*). However, the mean eccentricity was 9.0° for V4 in our data sets. How would V4 disparity-selective neurons with more foveal receptive fields (RFs) respond to our stimuli in terms of a match-based versus correlation-based representation of binocular disparity? To gain insights into this question, we ran simulations of a threshold energy model (*Doi and Fujita, 2014*; *Lippert and Wagner, 2001*). Although it is not a perfect explanation, adding a threshold (or expansive nonlinearity) to the stereo energy model is a well-supported mechanism that transforms the correlation-based representation into the match-

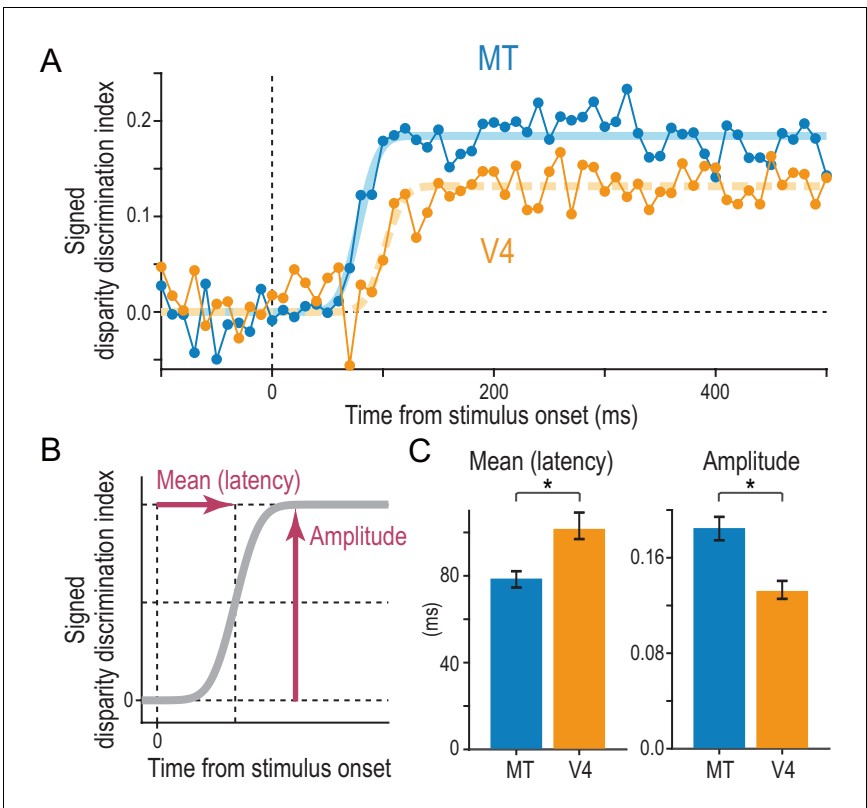

**Figure 8.** Comparison of the strength and speed of disparity selectivity between MT and V4 in response to cRDSs.
(**A**) The average time course of the signed disparity discrimination index (sDDI). The sDDI was calculated within 10 ms (non-overlapping) time windows for each cell and then averaged across cells separately for MT and V4 (N = 83 and 78, respectively). The smooth curves are fitted cumulative Gaussian functions. The vertical dashed line indicates the stimulus onset. (**B**) Parameters of the fitted cumulative Gaussian function. The amplitude parameter quantifies the saturation level of the sDDI. The mean parameter, our estimate of the selectivity latency, corresponds to the time required for the sDDI to rise to half the final amplitude. (**C**) Best-fitted values and 68% bootstrap confidence intervals. The latency and amplitude were significantly different between MT and V4.
The online version of this article includes the following source data for figure 8:

**Source data 1.** Time course of population-averaged signed disparity discrimination index.

based version (*Doi et al., 2013*; *Henriksen et al., 2016a*; *Henriksen et al., 2016b*; *Nieder and Wagner, 2001*; *Haefner and Cumming, 2008*). Furthermore, a combination of various subunits with additional nonlinearity could account for the characteristics of the disparity-tuning curves observed in V4 (*Abdolrahmani et al., 2016*). A fundamental feature of a foveal RF is its small size. Therefore, we examined the effects of RF size on the disparity representations of the threshold energy model as a proxy for the eccentricity effects. In simulations, we fixed the RF size (*Figure 9A*) and varied the stimulus scale (i.e., dot size) for convenience (*Figure 9B*), thus manipulating the relative size of RFs.

The response of the threshold energy model was more consistent with the match-based representation when the relative RF was smaller (i.e., when the eccentricity was smaller). *Figure 9C* shows the normalized disparity-tuning curves of the standard and threshold energy models at two extreme relative RF sizes. The responses of the standard energy model conformed to the pure correlation-based disparity representation, independent of the relative RF size of the model (*Figure 9C*, top row). (The tuning was wider for the smaller RF because the spatial auto-correlation of the monocular image was wider [i.e., the image had larger dots].)

The responses of the threshold energy model deviated from the pure correlation-based representation toward the match-based representation. Critically, the deviation was more prominent for smaller relative RFs. As in our analysis of the neuronal data, we drew the signed amplitude-ratio curves as a function of binocular correlation (*Figure 9D*) and computed the area ratios. The area

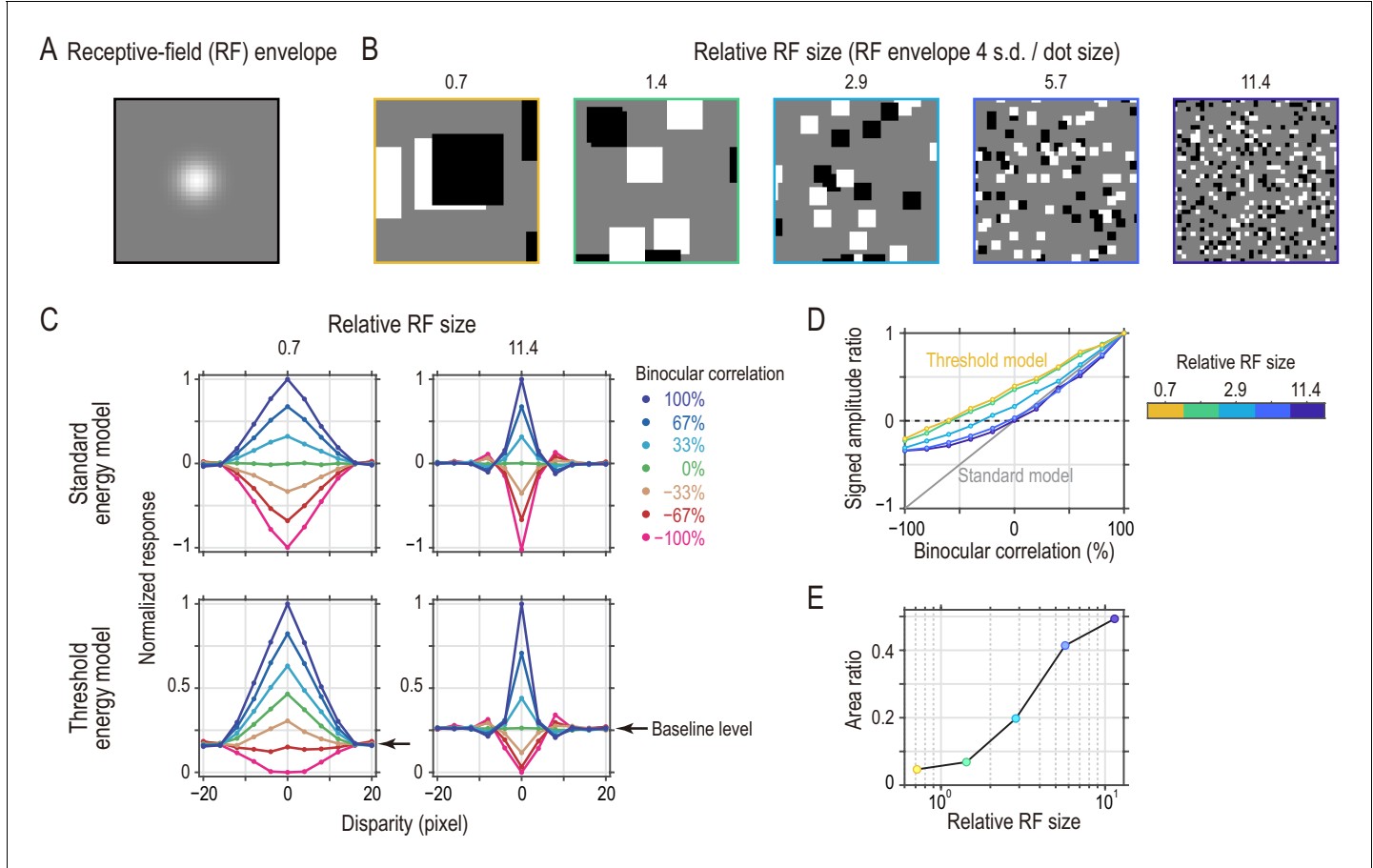

**Figure 9.** Threshold energy model with effectively smaller receptive fields (RFs) produces response patterns closer to match-based depth representation. (**A**) Gaussian envelope of a model RF (s.d., 2.86 pixels). We fixed the RF size in the simulations. (**B**) Random-dot stimuli with different spatial scales to simulate different relative RF sizes. The dot size ranged from 16 pixels to 1 pixel. (**C**) Disparity tuning curves from the standard energy model (top row) and those from the threshold energy model (bottom row). For the standard energy model, the response was defined as the binocular term of an energy-model complex unit. For the threshold energy model, these responses were half-wave rectified at each temporal frame. The tuning curves in each panel were normalized by the peak response (to 100% binocular correlation at zero disparity). Compare the simulated tuning curves to the pure correlation-based and match-based tuning shown in *Figure 1B*. (**D**) Signed-amplitude-ratio curves as a function of binocular correlation for different relative RFs. Compare the simulated results to the predictions shown in *Figure 4A*. The tuning amplitude was calculated as the tuning peak (or trough) minus tuning baseline. The negative sign indicates inverted tuning. (**E**) Area ratios computed for the curves in the threshold energy model in (**D**). The area ratio monotonically increased with the relative RF size.

The online version of this article includes the following source data and figure supplement(s) for figure 9:

**Source data 1.** Data for *Figure 9C* (disparity tuning).

**Source data 2.** Data for *Figure 9D* (signed-amplitude ratios).

**Source data 3.** Data for *Figure 9E* (area ratios).

**Figure supplement 1.** Distributions of instantaneous (frame-wise) responses of the standard energy model to half-matched RDSs.

**Figure supplement 1—source data 1.** Single-frame responses of the standard disparity energy model to half-matched random-dot stereograms.

ratio monotonically increased with the relative RF size, indicating that the models with smaller RFs represent binocular disparity in a more match-based manner compared to the models with larger RFs (*Figure 9E*). We discuss how the RF size affected the disparity representation in the supplementary materials (Appendix 2—Figure 9—text supplement 1, *Figure 9—figure supplement 1*).

Based on the simulation results presented above, we speculate that the disparity representation in V4 is more match-based for the neurons with foveal RFs than for those with peripheral RFs. Previous physiological studies (conducted in V1) provide data that are consistent with this view. As directly compared in *Goncalves and Welchman, 2017*, the reduction of the disparity selectivity for

aRDSs was more pronounced in *Cumming and Parker, 1997* than in *Samonds et al., 2013*. The former study sampled neurons over a range of eccentricity from 1° to 4°, whereas the latter concentrated on an eccentricity of 4°. (Their stimulus parameters were similar. The dot density was 25% for both studies. The dot size was 0.08° for the former and 0.094° for the latter.) We suggest that this eccentricity dependency on the disparity representation in V1, which was consistent with our simulation results, might be inherited in V4.

## Discussion

We used graded anticorrelation to identify whether MT and V4 neurons represented disparity based on binocular correlation (as prescribed by the disparity energy model; *Ohzawa et al., 1990*) or on binocular match (as consistent with the solution to the binocular correspondence problem) (*Figure 1*). An area-wise characterization of disparity tuning detected the signatures of the correlation-based representation in MT and those of the match-based representation in V4 (*Figure 3*). A finer, cell-by-cell characterization revealed a qualitative difference in the distribution of the representation type between the two areas (*Figure 5A–C*). MT might consist of two groups of neurons: one with the match-based representation and the other with the intermediate representation (*Figure 5A*). V4 lacked the intermediate group (*Figure 5B*). Importantly, in both areas, the shape of the disparity tuning (in responses to cRDSs) predicted the type of representation: more even-symmetric neurons showed stronger biases toward the match-based representation (*Figure 6*). This relationship was also observed for the inter-area comparison between MT and V4 (*Figures 5* and *7*). Lastly, although the type of disparity representation was more primitive overall in MT than in V4, the disparity signaling for cRDSs was stronger and faster in MT (*Figure 8*).

Differential disparity representations between MT and V4 have been suspected for more than a decade (*Parker, 2007*), but direct, convincing evidence has been lacking to date. Notably, the available MT and V4 studies with aRDSs used different statistical criteria (*Krug et al., 2004*; *Tanabe et al., 2004*). *Abdolrahmani et al., 2016* re-analyzed the MT data set in *Krug et al., 2004* and compared it against their V4 data set using the same statistical criterion, but failed to find a convincing difference between how the two areas responded to aRDSs (Abdolrahmani et al., Figure 11B). Consistent with this earlier result, we demonstrated that the disparity selectivity for aRDSs was indeed indistinguishable between MT and V4 when examined with the same methods (*Figure 5D–F*). Only the use of graded anticorrelation and a novel area-ratio metric revealed the MT-V4 difference in the disparity representation.

### Explanations for the links between even-symmetric disparity tuning and match-based disparity representation

The symmetry (phase) of disparity-tuning curves has been studied for decades to understand the neural coding of disparity (*DeAngelis et al., 1991*; *DeAngelis and Uka, 2003*; *Hinkle and Connor, 2005*; *Ohzawa et al., 1997*; *Poggio and Fischer, 1977*; *Poggio et al., 1988*; *Prince et al., 2002*; *Read and Cumming, 2004*; *Tanabe and Cumming, 2008*; *Tanabe et al., 2005*). Although many theories have been proposed (e.g., *Fleet et al., 1996*; *Goncalves and Welchman, 2017*; *Read and Cumming, 2007*), no empirical data have been reported regarding how the disparity-tuning symmetry relates to the process of solving a fundamental computational problem of stereopsis—the correspondence problem. In this study, we found that neurons with even-symmetric tuning, rather than those with odd-symmetric tuning, preferentially constituted the neural solution to the correspondence problem in the extrastriate areas MT and V4. More specifically, the disparity representations of even-symmetric neurons were more match-based, and thus more sophisticated, than the representations of their odd-symmetric counterparts in the middle stages of both dorsal and ventral pathways (*Figure 6*). No such relationships have been reported in any visual areas to our knowledge.

The observed link between disparity representation and tuning symmetry may reflect how the stereoscopic system has adapted to typical input images experienced in natural viewing conditions. Stereoscopic images are strongly constrained during natural viewing, because the two eyes see the same world from only slightly shifted vantage points. Naturally, the left- and right-eye images are strongly correlated, but are rarely anticorrelated. Primitive correlation-based mechanisms do not consider this natural constraint in the sense that they signal binocular disparity for anticorrelated images as strongly as for correlated images. By contrast, the match-based representation can be

considered the product of adaptation to naturally occurring binocular image pairs: the disparity selectivity for unnatural, anticorrelated images is selectively reduced (*Haefner and Cumming, 2008*). Why did this adaptation occur preferentially in neurons with even-symmetric disparity tuning? This is a sensible question given that simple feedforward mechanisms can reduce the disparity selectivity for anticorrelated stereograms in neurons with both even- and odd-symmetric tuning (*Read et al., 2002*; *Haefner and Cumming, 2008*).

The answer to this question may be related to the fact that due to their RF structures, even-symmetric neurons are more suited to represent natural binocular images than odd-symmetric neurons. The left- and right-eye RFs of even-symmetric neurons have the same shape except for a positional offset (*Ohzawa et al., 1997*; *Tanabe and Cumming, 2008*), mirroring the natural geometry of how an object in depth is projected to the two retinae with a positional offset. Conversely, odd-symmetric tuning implies that the underlying RFs have different shapes (phases) between the two eyes (for experimental verification in macaque V1 and V2, see *Tanabe and Cumming, 2008*; for energy model-based analyses, see *DeAngelis et al., 1991*; *Ohzawa et al., 1997*; *Prince et al., 2002*). Since an object's natural projections to the left and right eyes do not substantially differ in shape, these neurons do not maximally respond to natural binocular images. Indeed, an important role of odd-symmetric (and hybrids of odd- and even-symmetric) neurons in stereoscopic processing could be to detect the unnaturalness in binocular images (*Read and Cumming, 2007*; *Goncalves and Welchman, 2017*). Overall, the match-based representation may reflect the stereoscopic system's adaptation to naturally occurring pairs of binocular images (*Haefner and Cumming, 2008*). It seems reasonable that this adaptation occurs preferentially in neurons with even-symmetric disparity tuning, provided that one of their roles in stereopsis is to represent natural binocular images. Therefore, the even-symmetric neurons with the match-based representation can be viewed as dedicating their disparity encoding to naturally occurring stereoscopic images and thus efficiently coding binocular disparity (*Attneave, 1954*; *Barlow, 1961*).

## Contributions of the MT and V4 disparity signals to perception, eye movements, and downstream activity

MT and V4 had qualitatively different distributions of disparity representations (*Figure 5A–C*). Also, the disparity selectivity in V4 was, on average, about 30% slower and 30% weaker than that in MT, even though we used the random-dot stimuli tailored to drive V4 neurons. What could be the roles of each area in perception and action? We propose that MT pathway quickly and robustly relays the correlation-based disparity signals to the behavioral stage. The use of MT's disparity signals should not be limited to visually guided actions, because the link between the disparity-selective neurons in MT and perceptual judgment has been demonstrated extensively using a coarse depth task (*Chowdhury and DeAngelis, 2008*; *DeAngelis et al., 1998*; *Uka and DeAngelis, 2004*) and a structure-from-motion task (*Bradley et al., 1998*; *Dodd et al., 2001*; *Krug et al., 2004*). This implies that the correlation-based disparity signals in MT might contribute to some aspects of depth perception, even though the correspondence problem is not solved in these signals.

This idea is supported by our earlier psychophysical studies with the same graded anticorrelation paradigm (see *Fujita and Doi, 2016* for a review). Human depth judgment does indeed exhibit a trace of the correlation-based disparity representation (e.g., reversed depth perception for aRDSs; *Aoki et al., 2017*; *Tanabe et al., 2008*). Moreover, the inferred contribution of the correlation-based representation to perceived depth increases with the magnitude of stimulus disparity (*Doi et al., 2011*) as well as with the stimulus pattern refresh rate (*Doi et al., 2013*). The magnitude effect is consistent with the finding that MT contributes to coarse, but not fine, depth judgment (*Uka and DeAngelis, 2006*). The refresh-rate effect is consistent with the notion that MT is likely to contribute to perception more with dynamic stimuli than with static ones.

We also suggest that the disparity signals in MT drive the reflexive vergence eye movement via the medial superior temporal area (*Takemura et al., 2001*). For anticorrelation, the vergence eye movement responds in the opposite direction from stimulus disparity with a reduced amplitude (*Masson et al., 1997*), suggesting that the driving signal is intermediate between the match-based and correlation-based representations. Many MT neurons we observed fell into this category (see the second peak in the histogram in *Figure 5A*). The vergence response can be very fast, starting to rise only 60–70 ms after the stimulus onset (*Masson et al., 1997*). The fast and robust disparity signals that we observed in MT must be useful (*Figure 8*). Our suggestion is also consistent with the

finding that MT neurons represent the 'absolute' disparity of a stimulus, without being sensitive to the disparity of a reference plane (*Uka and DeAngelis, 2006*). This response property is a prerequisite for the vergence drive (*Westheimer and Mitchell, 1956*).

V4 is likely to be in the middle of the neural process of solving the binocular correspondence problem along the ventral pathway. Consistent with this idea is the finding that the correspondence problem is not yet fully solved in the single-neuron activity we observed in V4. The fraction of neurons with significantly disparity-selective tuning did not noticeably decrease from correlation levels 0% to −100% (*Figure 3B*), although the strength of the binocular match decreased from 50% to 0% over the same range (*Figure 1*). This discrepancy can be mitigated by aggregating the population responses just the way decision-making circuits would do (*Abdolrahmani et al., 2016*). This is because at −100% correlation, roughly half of the V4 neurons had inverted tuning curves (*Figure 3D*). If the responses of V4 neurons are aggregated according to their preferred disparity for cRDSs (i.e., the most naturalistic stimuli in our stimulus space), their tuning curves at −100% correlation are averaged out and flattened (*Doi et al., 2018*). This averaging out would not occur at 0% correlation, because the tuning shape at 0% correlation tended to be consistent with that at 100% correlation (as predicted by models of the match-based representation; *Doi and Fujita, 2014*; *Doi et al., 2013*). Thus, the aggregated V4 responses should resemble the pure match-based representation of disparity. This property of V4 neurons should presumably help downstream areas such as the inferior temporal (IT) cortex represent a complete solution to the correspondence problem at the individual-neuron level (*Janssen et al., 2003*).

We also suggest that V4 is critical for the perception of fine depth, because fine disparity perception entirely relies on the match-based representation (*Doi et al., 2011*). Several physiological results strongly support this view. First, V4 neurons have choice-predictive responses during a fine depth task (*Shiozaki et al., 2012*). Second, microstimulation of a cluster of V4 neurons biases the judgment of fine depth in the way predicted from the response property of the stimulated cluster (*Shiozaki et al., 2012*). Lastly, V4 neurons encode the 'relative' disparity between the center and surround of their RFs, a property necessary for supporting stereoacuity and a precursor for fixation-depth-invariant representation of 3D objects (*Umeda et al., 2007*).

V4 should also be important for 3D shape perception, because the perceptual discrimination of disparity-defined 3D shapes is difficult without match-based disparity signals (*Asher and Hibbard, 2018*; *Tanabe et al., 2008*). This is broadly consistent with the finding that V4 is involved in the encoding of 3D shape cues such as 3D slant (*Hegdé and Van Essen, 2005*; *Hinkle and Connor, 2002*). Overall, the signals in V4, including the selectivity for match-based disparity, fine disparity, and 3D shape cues, should help the ventral pathway derive the full-fledged representations of 3D objects and 3D scenes in its highest stage, the IT cortex (*Yamane et al., 2008*; *Verhoef et al., 2016*).

## Concluding remarks

We directly compared single-unit selectivity between dorsal and ventral visual areas using matched stimuli, following the footsteps of a few pioneering studies (*Cheng et al., 1994*; *Lehky and Sereno, 2007*). Specifically, we reported the division of labor between extrastriate areas MT and V4 in stereopsis, as revealed by a novel manipulation of RDSs. We suggest that area MT, a mid-tier stage in the dorsal pathway, relays the primitive correlation-based disparity signals without considerably refining the nature of the signals. Instead, this pathway has the advantage in terms of signaling speed and strength, which are useful for rapid vergence eye movements and coarse depth perception. By contrast, area V4, a counterpart stage in the ventral pathway, predominantly processes more sophisticated disparity signals based on binocular match. These signals help the end stage of the ventral pathway, the IT cortex, to fully solve the binocular correspondence problem and derive the fine representations of 3D objects. Therefore, the division of labor between the dorsal and ventral visual pathways may embody the tradeoff between the rapid and robust transmission of sensory signals and the complex computation needed to derive elaborate sensory representations.

This proposal provides a unified account for two earlier theories on the dorsal versus ventral division of labor (*Goodale and Milner, 1992*; *Ungerleider and Mishkin, 1982*). For example, vision related to action may depend more on the dorsal pathway because it benefits from rapid, robust sensory transmission, and so does visual motion perception. The analyses of colors and objects may depend more on the ventral pathway because they require complex computations.

We also note that the response characteristics of individual neurons are diversely distributed across MT and V4. Moreover, MT and V4 share a common link between the disparity-tuning shape and the type of disparity representation. It is possible that the interactions between the dorsal and ventral pathways play several roles in 3D visual perception and actions, as broadly suggested by neuroanatomy, single-neuron recording, electro-encephalography, psychophysics, and brain imaging (*Borra et al., 2008*; *Borra et al., 2010*; *Cottereau et al., 2014*; *Farivar, 2009*; *Fujita and Doi, 2016*; *Freud et al., 2016*; *Janssen et al., 2018*; *van Polanen and Davare, 2015*).

## Materials and methods

We recorded single neuron responses from areas MT and V4 of monkeys. Recordings in MT and V4 were conducted in separate sessions. All procedures were approved by the Animal Experiment Committee of Osaka University (permit numbers: FBS-12–016 and FBS-13-003-1) and conformed to the Guide for the Care and Use of Laboratory Animals issued by the National Institutes of Health, USA. The data sets are available online. Analysis codes for reproducing the results reported in this study are available upon reasonable request to the lead contact Ichiro Fujita (fujita@fbs.osaka-u.ac.jp). We previously analyzed the same data set from V4 (*Abdolrahmani et al., 2016*), but most of the analyses in the current paper are novel, with two exceptions (i.e., *Figure 2B,D* for an example neuron and *Figure 5E* for direct comparison of a conventional metric against our novel metric).

### Animal preparation

We used two Japanese monkeys (*Macaca fuscata*) for MT recording (Monkey O, male, weighing 8.5 kg; Monkey A, female, weighing 5.7 kg), and one Japanese monkey (Monkey O) and one rhesus monkey (Monkey I, *Macaca mulatta*, male, weighing 6.5 kg) for V4 recording. In these monkeys, a head-holding device was attached to the skull and Teflon-insulated stainless-steel search coils were implanted between the conjunctiva and sclera in both eyes under anesthetic and aseptic conditions. For details of the surgical procedures, see *Kumano et al., 2008*. Briefly, the monkeys were administered atropine sulfate (0.02 mg/kg, i.m.) to reduce salivation and were sedated with ketamine (5 mg/kg, i.m.). Anesthesia was maintained by inhalation of a mixture of isoflurane (0.3–2.0%, Forane, Abbott), nitrous oxide (66%), and oxygen (33%). Lidocaine was used for local anesthesia as needed. Throughout surgery, we continuously monitored electrocardiogram data, arterial oxygen saturation levels ($SPO_2$), end-tidal $CO_2$ levels, and heart rate. Postoperatively, we administered a general antibiotic (piperacillin sodium), ocular antibiotic (ofloxacin), glucocorticoid (betamethasone), and analgesic (ketoprofen). We trained the monkeys to perform a fixation task (see below). When the training was complete, we performed another surgery under the same anesthetic procedure to implant a chamber for recording (inner diameter, 19 mm; Crist Instruments, Hagerstown, MD). For MT recording, the chamber was placed on the occipital cortex 17 mm lateral and 14 mm dorsal to the occipital ridge tilted posteriorly at an angle of 25° above the horizontal. For V4 recording, it was placed 25 mm dorsal and 5 mm posterior to the external ear canals. The skull inside the chamber was removed with a trephine (Fine Science Tools, North Vancouver, Canada).

### Behavioral task

The monkeys viewed a CRT monitor (refresh rate 85 Hz; Multiscan G520, Sony) at a distance of 57 cm. The monitor covered a visual angle of 40° × 30°. We alternately presented images to the left and right eyes with custom-made OpenGL software, a quad-buffered graphics card, and shutter glasses based on ferroelectric liquid crystal devices (LV2500P-OEM, Citizen FineDevice, Yamanashi, Japan). The left and right shutter glasses alternately closed at each refresh of the display. Although the use of the shutter glasses decreases the effective binocular correlation compared with a haploscope, we have previously shown that the effective correlation is an unbiased, linear function of the notional correlation value (*Abdolrahmani et al., 2016*). Therefore, the use of shutter glasses did not complicate our interpretations as a result of artifactual distortion of the effective binocular correlation. Upon presentation of a white point at the center of the monitor, the monkeys were required to fixate it within 500 ms. After fixation, a visual stimulus was presented for 500 ms to both monkeys in the MT experiment and to monkey O in the V4 experiment; the stimulus duration was 700 ms for monkey I in the V4 experiment. The monkeys had to maintain fixation within a 1.5° × 1.5° window and a vergence angle within ±0.25° for another 200 ms. The eye positions were monitored with a

scleral eye-coil system (DSC-2000, Sankei Kizai, Tokyo, Japan). For successful trials, the monkeys received drops of juice or water as a reward. When the monkeys failed to maintain fixation, they did not receive a reward, and the data were discarded.

## Visual stimuli

We used circular patches of dynamic random-dot stereograms (RDSs) as visual stimuli. The RDSs consisted of a central disk and a surrounding ring with no gap between them. The central disk was positioned to cover the classical RF of a recorded neuron. The surrounding ring was 1.6° wide. RDSs contained an equal number of bright (2.28 cd/m$^2$) and dark (0.01 cd/m$^2$) dots with a gray background (1.14 cd/m$^2$). Luminance was measured through the shutter glasses placed between the monitor and a photometer (CS1000, Konica Minolta, Tokyo, Japan), and linearized (gamma corrected). We used only red phosphors to minimize inter-ocular crosstalk (<3% of the background) because the decay time is shorter for red phosphors than for green and blue phosphors. The dot size was 0.17° × 0.17° with anti-aliasing. The dot density was 25%.

We manipulated the binocular disparity and binocular correlation of the central disk, whereas these aspects of the surrounding ring were fixed at zero disparity and 100% correlation. We manipulated the binocular correlation of the central disk by reversing the luminance contrast of a varying proportion of dots in one eye (graded anticorrelation, *Figure 1A*; *Doi et al., 2011*; *Doi et al., 2013*). Note that this manipulation of the stimulus correlation is detectable only through binocular viewing; it does not change the overall luminance of the stimulus, because the RDSs are composed of the same numbers of bright and dark dots.

## Recording techniques

A tungsten electrode (0.3–2.0 MΩ at 1 kHz; FHC Bowdoinham, ME) was inserted into the cortex through a transdural guide tube using a micromanipulator (MO-971, Narishige, Tokyo, Japan). Extracellular neural signals were amplified (×5000; MDA4-I, Bak Electronics, Mount Airy, MD) and filtered (0.5–5 kHz; Multifunction Filter 3624, NF Corporation, Yokohama, Japan). Action potentials from single neurons were isolated by either a dual-window discriminator (DDIS-1, Bak Electronics, Mount Airy, MD) or a template-matching sorting system (Multi Spike Detector, Alpha-Omega Engineering, Nazareth, Israel). Times of spike occurrences and positions of both eyes were stored at 1 ms resolution for offline analysis.

We identified area MT based on the motion direction tuning of single- and multi-neuron activity, the relation between eccentricity and RF size, and changes in RF location along the electrode penetrations (*Gattass and Gross, 1981*; *Van Essen et al., 1981*), as well as general anatomical position. We confirmed the location of MT based on the pattern of gray matter and white matter encountered during electrode penetration before reaching MT as well as during the subsequent entry, after a short silent region (sulcus), into area MST, which had large, contralateral RFs covering a quadrant or half of the display (*Gu et al., 2006*; *Uka and DeAngelis, 2003*). After all recording experiments were finished, we made electrolytic lesions (10 nA for 10 s, electrode negative) at three to five sites in or around the recording regions. The monkeys were then euthanized with an overdose infusion of pentobarbital sodium (100 mg/kg, i.v.) and perfused from the heart with 0.1% phosphate-buffered saline (PBS) and 4% paraformaldehyde. The brains were removed, postfixed, frozen, and cut into 60-micron-thick serial, parasagittal sections. Alternate sections were stained for cell bodies with the Nissl method or for myelinated axons with the Gallyas method (*Gallyas, 1979*). We successfully recovered all the lesions. We verified that we recorded from MT deep in the posterior bank of a caudal part of the superior temporal sulcus where layers IV–VI contained a dense and uniform band of myelin fibers (*Ungerleider and Mishkin, 1979*; *Van Essen et al., 1981*). For the identification and RF information of V4 neurons, see *Abdolrahmani et al., 2016*.

Most of our MT neurons (65 out of 83) had their RFs in the lower visual field; RFs of all of our V4 neurons (N = 78) were located in lower visual field. The mean eccentricity was 6.9° (s.d. = 3.3°) for MT and 9.0° (s.d. = 3.0°) for V4 in our data sets. Based on the simulation and previous literature (*Cumming and Parker, 1997*; *Samonds et al., 2013*; *Goncalves and Welchman, 2017*), we speculate that disparity-selective neurons with smaller eccentricities might be associated with more strongly match-based disparity representations. If this speculation is correct, we might be slightly underestimating the representational difference between MT and V4. Alternatively, the slight mean

eccentricity difference between the two areas might not have affected our conclusion given that the standard deviation of the eccentricity in each area was quite large compared to the mean difference. Our data appear more consistent with the latter view. We found no correlation between the area ratio (our metric for disparity representation) and eccentricity either in MT (Spearman's correlation $r_s$ = −0.035, p=0.79) or in V4 ($r_s$ = −0.065, p=0.68). The results did not change even when the reflected symmetry phase was controlled (partial Spearman's correlation $r_s$ = −0.055, p=0.68 for MT; $r_s$ = −0.043, p=0.79 for V4). We do not think that the lack of significant correlation convincingly falsified the speculated relationship between eccentricity and disparity representation, because there could be many reasons why the relationship was not detected (e.g., the eccentricity variation was not large enough to detect the relationship; note that in the simulation the size of the largest RF examined was 16 times that of the smallest RF).

## Experimental procedures

Here we describe the procedures used for area MT. We used similar procedures for area V4, but see *Abdolrahmani et al., 2016* for details. After isolating single-neuron activity, we determined its classical RF by a patch of RDSs (size: 3° or smaller, disparity: 0°, dot-pattern refresh rate: 10.6 Hz in most cases). We measured direction tuning with a patch of dots coherently moving in one of eight directions (45° apart). We then probed the range of disparity tuning with dynamic, non-motion-coherent RDSs in several (at least three) stimulus presentation trials. For this initial test, we used cRDSs with nine disparities (−1.6° to 1.6° in 0.4° intervals) and uncorrelated RDSs (uRDSs; RDSs with independently created left-eye and right-eye images). After all these tests, we measured the responses to RDSs with several binocular disparities (tailored for each recorded neuron based on the initial test) and correlation levels (100%, 70%, 30%, 0%, −30%, −70%, and −100%). The dot-pattern refresh rate was set to 10.6 Hz, the value used for the V4 experiment.

## Data analysis

All analyses were carried out with MATLAB (The MathWorks). The spontaneous firing rate was calculated for a 250 ms period preceding the stimulus onset and averaged across trials. To construct disparity-tuning curves, the mean firing rate at each correlation level and disparity was calculated for a certain time window. The time window overlapped with the stimulus presentation period but was delayed to take into account response onset and offset latency (60 ms for MT and 80 ms for V4). The average firing rate was based on at least five trials. One of the monkeys (Monkey O) was used in both MT and V4 experiments. The V4 data were used in our previous publication (*Abdolrahmani et al., 2016*), but the main analyses of the current paper are all new. Particularly, we quantified the disparity representation of each V4 neuron and directly compared it against MT results in this study.

### Gabor function for disparity tuning

We characterized neuronal disparity tuning by fitting a Gabor function, a product of a Gaussian envelope and a cosine carrier, to the mean firing rate data. At each level of binocular correlation, the mean firing rate ($R$) at disparity ($x$) was defined as follows:

$$R(x) = y_0 + A \cdot e^{\left(\frac{-(x - x_0)^2}{2\sigma^2}\right)} \cdot \cos[2\pi f(x - x_0) + \varphi],$$

where $A$, $y_0$, $x_0$, $\sigma$, $f$, and $\varphi$ represent the amplitude, baseline response, horizontal position, envelope width, carrier frequency, and carrier phase, respectively. We performed a half-wave rectification for $R$ because the firing rate cannot be negative. Only the amplitude and phase parameters were independently fitted across different binocular correlation levels, because the differences in these two parameters characterize the correlation-based and match-based representations (*Figure 1B*). The other parameters were shared across different correlation levels. The fmincon function in the MATLAB Optimization Toolbox was used for finding the parameter combination that minimizes the summed squared error between the single-trial firing rates and Gabor functions. The goodness of fit was estimated by taking the $R^2$ between the mean firing rates and the best-fitted Gabor function separately for each correlation level. After fitting, we selected 76 MT and 71 V4 disparity-selective neurons that had good fitting quality for the responses to cRDSs ($R^2$ >0.6).

## Signed amplitude ratio

To characterize how a decrease in binocular correlation from 100% changes disparity tuning, we devised the signed amplitude ratio by modifying a conventional metric, the amplitude ratio (*Cumming and Parker, 1997*). We computed the signed amplitude ratio at each level of binocular correlation for each cell. The conventional amplitude ratio quantifies how much the fitted Gabor amplitude changes because of the anticorrelation of stereograms. The conventional metric was given a sign that was determined based on how anticorrelation changed the shape of disparity tuning. When anticorrelation inverted the shape, we gave a negative sign. We detected the shape inversion based on the change in the symmetry phase (see below). The symmetry phase difference between two tuning curves had to be smaller than $-0.5\pi$ or larger than $0.5\pi$ for the shape inversion to be detected.

## Reflected symmetry phase

When the cycle of the carrier sinusoid is much larger than the Gaussian envelope width, the Gabor phase may not be a good indicator for the true shape of disparity-tuning curves (*Read and Cumming, 2004*). In this case, the sinusoidal carrier does not constrain the shape of the fitted Gabor function. Accordingly, the neurons with even-symmetric tuning curves can be classified as odd symmetric if we use the fitted value of the carrier phase (confirmed in our data set). We therefore characterized the shape of the disparity-tuning curve by calculating the symmetry phase (*Read and Cumming, 2004*). This metric directly estimates the relative weights of even-symmetric and odd-symmetric components in the overall shape of the fitted tuning curve. The symmetry phase of zero or $\pi$ corresponds to the pure even symmetry, whereas the symmetry phase of $\pm \pi/2$ corresponds to pure odd symmetry. To construct a metric that monotonically quantifies the tuning symmetry, we devised reflected symmetry phase. We changed the symmetry phase to the absolute value. Then, we wrapped the absolute value into the range from zero (purely even symmetric) to $\pi/2$ (purely odd symmetric) by reflecting it at $\pi/2$.

## Area ratio

We characterized how well each of the recorded disparity-selective neurons followed the correlation-based or match-based prediction using their responses over the entire range of binocular correlation. This analysis consisted of two steps. First, we characterized how the signed amplitude ratio changed as a function of binocular correlation. Although the simplest examples shown in *Figure 4A* are linear functions, we fitted a quadratic function to the observed data and this often showed nonlinear trends (*Figure 4C,D*). The quadratic coefficient was almost always positive, meaning that the nonlinearity was expansive (84% for MT, 90% for V4). We quantified the degree of nonlinearity using the ratio of the quadratic coefficient to the linear coefficient from the best-fitted quadratic function (*Britten et al., 1993*). The ratio was significantly higher than zero for both MT and V4 (MT: median, 0.02; lower-quartile, 0.0057; upper-quartile, 0.0597; two-sided Wilcoxson signed-rank test, p=3.6 × 10⁻¹⁴; V4: median, 0.02; lower-quartile, 0.0116; upper-quartile, 0.028; p=2.4 × 10⁻¹³), validating our use of quadratic functions. The function was constrained to take the value of one at 100% correlation. The sum-squared error was minimized. Second, we calculated the area ratio to quantify how well the fitted quadratic function followed the correlation-based prediction versus the match-based prediction (*Figure 4A,B*). The denominator of the area ratio was the area (integral) of the fitted quadratic function across the range where the fitted function was positive (gray triangle in *Figure 4B*). The numerator was the area of the function over the range where the function was negative (blue triangle). Because the correlation-based representation should have perfectly inverted disparity-tuning curves for two binocular correlation values with the same magnitude but opposite signs, it should lead to an area ratio of one. By contrast, the match-based representation should have a flat tuning curve at −100% correlation, leading to an area ratio of zero.

## Signed disparity discrimination index

The conventional DDI measures the disparity discriminability of a neuron as follows (*Prince et al., 2002*):

$$DDI = \frac{R_{pref} - R_{null}}{R_{pref} - R_{null} + 2\sqrt{SSE/(N-M)}} \, ,$$

where $R_{pref}$ and $R_{null}$ indicate the mean firing rates of a neuron at its preferred and null disparities, respectively (in response to cRDSs). The SSE is the sum of the squared error of single-trial responses (around the mean; that is, the tuning curve); $N$ and $M$ indicate the total number of trials and the number of tested disparities, respectively. A DDI of 0 indicates no selectivity (i.e., a completely flat tuning curve); a DDI of 1 indicates that the tuning curve amplitude is so large that the noise level is negligible.

We wanted to compute DDIs using small, sliding time windows to examine how the disparity selectivity developed in MT and V4. Therefore, we modified the conventional DDI as follows:

$$sDDI(t) = \frac{R_{pref}(t) - R_{null}(t)}{\left| R_{pref}(t) - R_{null}(t) \right| + 2\sqrt{SSE(t)/(N-M)}} \, ,$$

where $t$ indicates the time from the stimulus onset. The firing rates were computed over the window $\pm 5$ ms around the time $t$. Note that we defined the preferred and null disparities based on the responses over the entire stimulus duration (as for the standard DDI), because our aim was to quantify the instantaneous selectivity that was congruent to the overall selectivity. If we defined the preferred and null disparities at each time $t$, the sDDI value would always be above zero (due to the circular logic), complicating the interpretation. With our definition, the sDDI could have a negative value when the selectivity at time $t$ differed from the selectivity over the entire stimulus duration. This tended to happen more often when $t$ was small, and well before the cell's response latency. During this period, responses are noisy and the sign of the sDDI is stochastic.

## Cumulative Gaussian function for the sDDI time course

We fitted the cumulative Gaussian function with an amplitude parameter (*Figure 8B*) to the sDDI time course for MT and V4. During the fitting, we minimized the sum of the squared errors between the fitted function and the mean sDDI (averaged across cells in each area). We used bootstrapping to estimate the 68% confidence intervals of fitted parameter values. When constructing an artificial data set, we resampled the neurons with replacement for each area (N = 83 and 78 for MT and V4, respectively) and computed the new sDDI mean value at each time point. The confidence interval was computed based on 10,000 resampled data sets.

## Computation of the area ratio without model fitting

We computed the area ratio entirely without model fitting and confirmed the same conclusion. For this model-free analysis, we substituted the amplitude parameter of the fitted Gabor function with the peak-to-trough difference of raw disparity-tuning data. The sign of the amplitude ratio was determined using the sign of the correlation coefficient between the raw tuning data at 100% binocular correlation and those at lower binocular correlations. This was because the tuning-curve inversion should lead to a negative correlation coefficient. We also substituted the quadratic-function fitting to signed amplitude-ratio data with the piecewise linear interpolations between two adjacent raw data points. Then, we calculated the area ratio based on the areas surrounded by these piecewise linear interpolations and the horizontal baseline (zero signed amplitude ratio). The numerator and denominator of the area ratio were the net areas below and above baseline, respectively. For the model-free analysis, we used our entire data set (83 MT and 78 V4 disparity-selective neurons).

## Simulation

We simulated the responses of the standard disparity energy model (*Ohzawa et al., 1990*) and the threshold energy model (*Doi and Fujita, 2014*; *Lippert and Wagner, 2001*). For the standard energy model, we subtracted out the monocular response terms to isolate the binocular response term of a complex cell. For the threshold energy model, single-frame responses from the standard model were half-wave rectified and then temporally averaged. Subtraction of the monocular terms before subjecting model responses to output nonlinearity was performed in previous studies (*Thomas et al., 2002*; *Doi and Fujita, 2014*). For the details of the energy model, see *Doi et al.,*

*2013* (Experiment 3). Briefly, the model parameter values we used were: Gaussian envelope sigma (for both vertical and horizontal orientations), 0.05°; sinusoidal carrier frequency, three cycles/°. Both the position and phase RF disparities were zero.

Our goal was to examine the effects of RF size on the model responses. The most convenient way to simulate the change in RF size was to fix the size and vary the spatial scale of the random dot stereograms (RDSs), because RDSs consisted of discrete dots. We varied the dot size from 1 pixel × 1 pixel to 16 pixels × 16 pixels (0.0175°/pixel). This corresponded to relative RF sizes ranging from 0.7 to 11.4 (the relative RF size was defined as 4 × the RF envelope's standard deviation relative to the dot size). Dot density was also varied to maintain ~25% of non-background pixels (if the dots had not overlapped). Both RFs and stimuli were defined as 37 pixels × 37 pixels. A single trial consisted of 64 frames (i.e., independent dot patterns). We simulated the average response over 100 trials in each condition.

For disparity-tuning curves, we simulated 11 disparity values from −20 to 20 pixels and seven binocular correlation levels from −100% to 100% (*Figure 9C*). For the signed amplitude ratio, we tested three disparity values (−20, 0, and 20) and 11 correlation levels from −100% to 100% (*Figure 9D*). The tuning peak or trough was computed based on the responses to zero disparity, while the tuning baseline was computed based on the responses to disparity values of −20 and 20 pixels. We linearly interpolated discrete data points in *Figure 9D* to compute the area ratio. *Figure 9—figure supplement 1A* shows the distributions of instantaneous responses (i.e., responses to individual dot patterns before temporal averaging). Each distribution was based on 64 frames × 100 trials.

## Acknowledgements

We thank H Ban, Y Sakano, HM Shiozaki, and BG Wundari for comments on earlier versions of the manuscript; S Mita, T Oga, M Onoue, and M Inagaki for technical assistance; T Uka, K Okada, Y Kobayashi, H Kumano, TM Sanada, and M Saruwatari for technical advice; and M Omokawa for animal care. This work was supported by grants to IF from the Ministry of Education, Culture, Sports, Science and Technology of Japan (MEXT; 23240047, 15H01437, 17H01381, 18H05007) and the Ministry of Internal Affairs and Communications. The monkeys were provided by the National Institute of Natural Sciences (NINS) through the National Bio-Resource Project (NBRP) of the MEXT, Japan.

## Additional information

### Funding

| Funder | Grant reference number | Author |
|---|---|---|
| Ministry of Education, Culture, Sports, Science and Technology | 2324007 | Ichiro Fujita |
| Ministry of Education, Culture, Sports, Science and Technology | 15H01437 | Ichiro Fujita |
| Ministry of Education, Culture, Sports, Science and Technology | 17H01381 | Ichiro Fujita |
| Ministry of Education, Culture, Sports, Science and Technology | 18H05007 | Ichiro Fujita |
| Ministry of Internal Affairs and Communications | | Ichiro Fujita |

The funders had no role in study design, data collection and interpretation, or the decision to submit the work for publication.

### Author contributions

Toshihide W Yoshioka, Conceptualization, Data curation, Software, Formal analysis, Validation, Investigation, Visualization, Writing - original draft, Writing - review and editing; Takahiro Doi,

Conceptualization, Data curation, Software, Formal analysis, Validation, Visualization, Writing - original draft, Writing - review and editing; Mohammad Abdolrahmani, Conceptualization, Data curation, Software, Formal analysis, Validation, Investigation, Writing - original draft, Writing - review and editing; Ichiro Fujita, Conceptualization, Resources, Supervision, Funding acquisition, Writing - original draft, Project administration, Writing - review and editing

#### Author ORCIDs
Toshihide W Yoshioka https://orcid.org/0000-0001-6475-4627
Takahiro Doi https://orcid.org/0000-0002-9650-972X
Mohammad Abdolrahmani https://orcid.org/0000-0003-3658-2623
Ichiro Fujita https://orcid.org/0000-0003-3293-8610

#### Ethics
Animal experimentation: All experimental procedures in this study were approved by the Animal Experiment Committee (Permit Numbers: FBS-12-016, FBS-13-003-1) of Osaka University, and conformed to the Guide for the Care and Use of Laboratory Animals issued by the National Institutes of Health, USA.

#### Decision letter and Author response
Decision letter https://doi.org/10.7554/eLife.58749.sa1
Author response https://doi.org/10.7554/eLife.58749.sa2

## Additional files

#### Supplementary files
• Transparent reporting form

#### Data availability
Source data files have been provided for all data figures.

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

## Appendix 1

### Figure 5—text supplement 1
Neurons with non-monotonic signed amplitude-ratio functions

Not all neurons in our data sets allowed us to readily interpret their activity in terms of our theoretical framework, because some neurons did not fall into the spectrum between correlation-based and match-based representations. Specifically, neither type of representation predicted the non-monotonic change in the signed amplitude ratio. Strictly speaking, any non-monotonic change, unless caused by noise, was a violation of our framework, but we categorized observed patterns into two kinds: weak and strong violations.

For the weak violation, the signed amplitude-ratio function was non-monotonic but either crossed the horizontal baseline (i.e., zero) only once or always stayed positive. For these data patterns, the area ratio still works reasonably well. As the area ratio increases, the signed amplitude-ratio function becomes closer to the correlation-based prediction.

For the strong violation, the signed amplitude-ratio function crossed the horizontal baseline more than once. For example, the function could have a positive value at −100% correlation and the negative minimum at a correlation level above −100% (see *Figure 5—figure supplement 4A* for single-neuron examples). When individual data points were used for the assessment, the strong violation was found in 21% of the disparity-selective neurons in MT (16/76) and 32% of those in V4 (23/71) (see *Figure 5—figure supplement 4B* for the average pattern of the function for these neurons). These percentages might be overestimated by somewhat noisy nature of individual data points. When well-fitted curves were considered (i.e., quadratic functions with $R^2$ >0.6), the percentages dropped to 7% in MT (4/59) and 9% in V4 (4/43). These neurons were included in the model-free analysis presented in *Figure 5—figure supplement 3A* (see Materials and methods for how the area ratio was calculated). We removed these neurons and confirmed the same conclusion: the median area ratio was different between MT and V4 (*Figure 5—figure supplement 4C*).

## Appendix 2

### Figure 9—text supplement 1
Receptive-field (RF) size effects on disparity representation in a threshold energy model

Why does a small RF accentuate the role of threshold nonlinearity in transforming energy-model responses into a match-based representation? More specifically, why does the threshold energy model with a small RF produce more profound disparity tuning for half-matched random-dot stereograms (hmRDSs; i.e., 50% binocular correlation) and weaker tuning for anticorrelated RDSs (aRDSs; i. e., −100% binocular correlation) compared to the same model with a larger RF? In an attempt to answer these questions, we visualized the response distributions of the standard energy models for hmRDSs (*Figure 9—figure supplement 1A*). We analyzed instantaneous responses, that is, responses to individual temporal frames (dot patterns) that have yet to go through output nonlinearity. The response was normalized in the same way as in *Figure 9C* and compared between the two extreme RF sizes and between two disparity values (0 pixels as the preferred disparity and −20 pixels as the baseline disparity).

First, why was the disparity tuning of the threshold model for hmRDSs more profound when the RF was smaller? We found that when the RF was small, the response distribution (of the standard model) differed between the preferred and baseline disparities (*Figure 9—figure supplement 1A*, left column). At the preferred (zero) disparity, the responses more often deviated from zero. In the threshold model, this deviation was transformed into a positive mean response (vertical dashed line) because half-wave rectification was applied before temporal response integration. (In the standard model, the bidirectional deviation was simply averaged out, hence there was no disparity selectivity for hmRDSs.) When the RF was large, the response distribution did not differ between the two disparity values (*Figure 9—figure supplement 1A*, right column). At the preferred disparity, positive signals from contrast-matched dots and negative signals from contrast-reversed dots canceled each other out in the large RF in every temporal frame.

Second, why was the disparity tuning of the threshold model for aRDSs weaker when the RF was smaller? We noticed that the tuning baseline was lower for the smaller RF (compare the arrow heights between the bottom left and bottom right panels in *Figure 9C*). To understand the baseline difference of the threshold model between the small and large RFs, we looked at the response distributions of the standard model at the baseline disparity (*Figure 9—figure supplement 1A*, top row; note that for the baseline disparity, stimulus binocular correlation no longer matters, and therefore the responses to hmRDSs can be used as a surrogate for the responses to aRDSs). The response distribution was closer to zero for the small RF than for the large RF. This led to the reduced baseline response of the threshold model with the smaller RF.

