## [Decision Letter]

**Acceptance summary:**

There are very few studies that use a well-controlled stimulus paradigm to quantitatively compare responses in the dorsal and ventral visual pathways. By using a single anti-correlated random dot stimulus, this elegant paradigm nicely illustrates the match-based behavior of V4 neurons vs the correlation-based behavior of MT neurons. In addition, despite the stimulus being a non-moving RDS stimulus, V4 and MT are clearly distinguished by response strength and latency. This study makes a significant contribution to the field by solidifying dorsal/ventral pathway distinctions in disparity processing and identifies a key perceptually relevant test that unequivocally and quantitatively distinguishes stereoscopic function in these two pathways.

**Decision letter after peer review:**

Thank you for submitting your article "Specialized contributions of mid-tier stages of dorsal and ventral pathways to stereoscopic processing in macaque" for consideration by *eLife*. Your article has been reviewed by three peer reviewers, including Anna Wang Roe as the Reviewing Editor and Reviewer #1, and the evaluation has been overseen by Chris Baker as the Senior Editor. The following individual involved in review of your submission has agreed to reveal their identity: Andrew J Parker (Reviewer #2).

The reviewers have discussed the reviews with one another and the Reviewing Editor has drafted this decision to help you prepare a revised submission.

Summary:

This is an interesting paper that contains a very valuable comparison of neuronal performance in binocular vision between two visual areas, one (V5/MT) in the dorsal stream and the other (V4) in the ventral stream. The paper builds upon a number of previous findings from this research group and others using the technique of testing binocular visual neurons with inputs that vary from being strongly correlated between left and right eyes' images (random dot stereograms, RDS) to complete anti-correlation (anti-random dot stereograms, aRDS). The prediction is that neurons in V4 behave in a match-based way and MT in a correlation-based way (aRDS produces inversion in curve correlation). The data support this prediction well. That is, V4 neurons show greatly reduced “negative part” of the tuning curve, consistent with a non-linear exclusion operation. MT neurons, in contrast, exhibit robust negative parts of the tuning curves, similar to responses of V1 neurons. The results suggest significant differences in the visual processing of binocular information in the two areas.

The results compare the response to two different ways of manipulating the binocular correlation structure of images. The first varies from correlated (+1) to anticorrelated (-1) by replacing some dot pairs in matched locations in the left and right eyes with dots placed at randomly chosen locations in the left and right eyes. With this manipulation, as dots in matched locations are replaced by dots in randomly chosen locations, the correlation level of two images falls away from +1 and -1, eventually approaching 0 when all dots are in randomly chosen locations. In the second manipulation, dots remain in their matched locations but the proportion of them that are correlated or anticorrelated is varied. This manipulation arrives at an interesting combination of dots which is half correlated and half anticorrelated: the average correlation of such a figure is zero, but the structural distribution of correlations is different from the previous manipulation in which randomly chosen dot locations are applied. The authors exploit this and call this second stimulus half-matched, because half of the dots have positive correlation and the signal from these dots leads to a perceptual sense of depth, whereas there is no specific depth apparent when the images are binocularly uncorrelated by placing dots in random locations.

For both these types of stimulus with an average of zero correlation, the response of an idealized detector of binocular correlation (such as the Ohzawa, DeAngelis and Freeman model) would naturally be zero in both cases. However, as the authors recognize, the slightest non-linearity in neural processing would lead to different responses to these two stimuli. Therefore, the authors have devised a way of assessing the responses to these two stimuli across the whole range of correlation values. This is helpful and presents some interesting findings.

Essential revisions:

1) Area-ratio measure: A major concern is that not every aspect of the data will have been captured by the new area-ratio measure. The issue is the relationship between signed-amplitude ratio and binocular correlation (as sketched in Figure 4A of the submitted paper). The assumption of the paper is that this is always monotonic as a function of correlation, but there are already clear published examples where this is not true (Figure 1A of Krug et al., 2004 is one example). In this case, the relationship between signed-amplitude ratio and binocular correlation would be V-shaped and nonlinear, not monotonic and linear as shown in Figure 4A). So, the empirical question would be whether the data-sets for this paper contain any signs of this non-linear relationship.

2) Symmetry: (a) The main finding here is that the disparity tuning symmetry is correlated with the disparity representation in both MT and V4. More specifically, neurons with even-symmetric tuning tended to have match-based representations and neurons with odd-symmetric tuning tended to have correlation-based representations. Although the correlation was rather robust in MT (r = 0.55), in V4 this correlation was weak (r = 0.34). The authors should acknowledge the fact that tuning symmetry could only explain 11% of the variance in the disparity representation metric (the area ratio). Maybe the correlation coefficient did not capture the nonlinear relationship between the two metrics in V4, and therefore other ways to quantify this relationship should be considered. (b) The authors spend a section in the Discussion on response symmetry. However, it is not clear what the functional significance of this finding is (beyond the obvious). What new offered by the symmetry finding? "neurons with even-symmetric tuning preferentially constituted the neural solution to the correspondence problem in extrastriate areas." Is this surprising? Somehow the message did not come through.

3) Novelty and significance: Precisely what new has been learned here? This must be spelled out clearly. One gets the impression that the main observations in this study are not new. The authors themselves published work on V4 (some of these data were used in this study), and Krug et al., 2003 described the responses of MT neurons to anticorrelated disparity. (Possibly the only difference being the MT neurons with an area ratio of 1, which were described in Krug et al. but were almost absent in this study.) Perhaps the most novel aspect of this study is the use of traditional ventral pathway stimulus (static RDS and aRDS stimuli) for dorsal pathway study. These results (fast robust response) serve to further support the view of MT/dorsal pathway in action vision. The V4 findings, while well conducted, do not significantly change or extend our view of V4 function in binocular disparity.

4) Overstated conclusions: (a) The conclusion that the data demonstrate “a general principle of the stereoscopic system both within and across the dorsal and ventral pathways” seems too strong based on this study in only two areas. For this conclusion, we need data in many other areas. Nevertheless, I also acknowledge that this will not be easy because most visual areas contain either a strongly match-based representation (such as inferotemporal cortex and parietal areas AIP), or a correlation-based representation such as V1 and V2. (b) Similarly, the conclusion that “neurons with even-symmetric tuning preferentially constituted the neural solution to the correspondence problem in extrastriate areas” seems too general. (c) The authors statement that “the correlation-based disparity signals in MT directly contribute to some aspects of depth perception” is too strong since the behavioral effects of microstimulation in MT could have been caused by the match-based neurons alone (since microstimulation does not discriminate between the two types of neurons).

5) Visual eccentricities of recordings in V4 and MT: The mean eccentricities of the V4 neurons are ~9 deg while that of MT neurons are ~7 deg. This seems rather eccentric for V4. Do these findings hold for RFs in more central (0-3 deg) eccentricity? This is important as V4's primary function is in central vision.

---

## [Author Response]

Essential revisions:1) Area-ratio measure: A major concern is that not every aspect of the data will have been captured by the new area-ratio measure. The issue is the relationship between signed-amplitude ratio and binocular correlation (as sketched in Figure 4A of the submitted paper). The assumption of the paper is that this is always monotonic as a function of correlation, but there are already clear published examples where this is not true (Figure 1A of Krug et al., 2004 is one example). In this case, the relationship between signed-amplitude ratio and binocular correlation would be V-shaped and nonlinear, not monotonic and linear as shown in Figure 4A). So, the empirical question would be whether the data-sets for this paper contain any signs of this non-linear relationship.

The reviewers are correct in stating that our data sets contained some neurons that were difficult to readily interpret within our framework. We have now characterized these neurons, removed them in a new control analysis, and confirmed the original conclusion. The results are presented in Figure 5—figure supplement 4 and Appendix 1—Figure 5—text supplement 1.

As for the comparison to Krug et al., 2004, the phase-difference distribution for completely anticorrelated RDS (aRDS) in our MT data set did not appear very different from Figure 3C of Krug et al., 2004 (Author response image 1). (Note that their Figure 3 shows the results from planar RDSs, in which the stimuli are relatively comparable to those of our RDSs, while Figure 1 showed the results from cylinder stereograms, which are rather different from our stereograms.) Our V4 data showed a noticeable peak at zero, indicating the presence of neurons with mostly non-significant but similarly shaped tuning for aRDSs as compared to the tuning for cRDSs. We presented detailed analyses of V4 responses in Abdolrahmani et al., 2016.

**Author response image 1. sa2fig1:** Distributions of symmetry-phase differences for aRDSs (i.e., at −100% binocular correlation). Solid bars indicate neurons with significant disparity selectivity for aRDSs (Kruskal-Wallis test, p < 0.05). The histogram for V4 is a modified plot of Figure 7A (bottom left) in Abdolrahmani et al., 2016. N = 76 and 71 for MT and V4, respectively (neurons with well-fitted Gabor tuning curves, R^2^ > 0.6).

“Third, we removed the neurons with particular types of non-monotonic, signed-amplitude ratio functions that could complicate the interpretation of area ratio (Figure 5—figure supplement 4, Appendix 1—Figure 5—text supplement 1).”

2) Symmetry: (a) The main finding here is that the disparity tuning symmetry is correlated with the disparity representation in both MT and V4. More specifically, neurons with even-symmetric tuning tended to have match-based representations and neurons with odd-symmetric tuning tended to have correlation-based representations. Although the correlation was rather robust in MT (r = 0.55), in V4 this correlation was weak (r = 0.34). The authors should acknowledge the fact that tuning symmetry could only explain 11% of the variance in the disparity representation metric (the area ratio). Maybe the correlation coefficient did not capture the nonlinear relationship between the two metrics in V4, and therefore other ways to quantify this relationship should be considered.

In the revised manuscript we have explicitly acknowledged that the observed correlation in V4 was moderate (normally a correlation value above 0.3 is described as moderate rather than weak). We added a supplementary analysis with linear and quadratic regressions (Figure 6—figure supplement 1). In V4, R^2^ was 0.20 for the linear regression and 0.31 for the quadratic regression. (Note that R^2^ for the linear regression was not 0.11 because we used Spearman’s rank correlation.) We apologize for the fact that there was a mistake in the correlation value for MT; the correct value is r_s_ = 0.41, p = 1.4 × 10^−3^.

“Note that the observed correlation was only moderate for both MT and V4. The best-fitted linear model accounted for 24% of the total variance in the area ratio for MT and 20% for V4 (Figure 6—figure supplement 2A, top row). These percentages increased only to 25% and 31% with the quadratic model in MT and V4, respectively (Figure 6—figure supplement 2A, bottom row). These results imply that the tuning symmetry is only part of the explanation for the cell-by-cell variation in disparity representations. Consistent with this view, the width of the spatial-frequency tuning is correlated with the amplitude ratio in V4 (Kumano et al., 2008).”

(b) The authors spend a section in the Discussion on response symmetry. However, it is not clear what the functional significance of this finding is (beyond the obvious). What new offered by the symmetry finding? "neurons with even-symmetric tuning preferentially constituted the neural solution to the correspondence problem in extrastriate areas." Is this surprising? Somehow the message did not come through.

We are sorry that our original manuscript did not clearly convey the novelty and significance of our findings on the tuning symmetry.

Novelty:

As the reviewers noted below, the following statement was too general: "neurons with even-symmetric tuning preferentially constituted the neural solution to the correspondence problem in extrastriate areas." We just tried to summarize the results shown in Figure 6, which are clearly and empirically novel since similar relationships have not been previously reported in any visual areas as far as we are aware. We revised the text to clarify and emphasize our message.

“… neurons with even-symmetric tuning, rather than those with odd-symmetric tuning, preferentially constituted the neural solution to the correspondence problem in the extrastriate areas MT and V4. More specifically, the disparity representations of even-symmetric neurons were more match-based, and thus more sophisticated, than the representations of their odd-symmetric counterparts in the middle stages of both dorsal and ventral pathways (Figure 6). No such relationships have been reported in any visual areas to our knowledge.”

Significance:

The main significance of the finding is that we can understand the link between match-based representation and even-symmetric tuning by considering naturally occurring pairs of binocular images. To clarify this point, we deleted the second significant point (mechanism), moved the related paragraph from the Results section, and thoroughly revised the remaining text. We hope that our message is now clear.

“The observed link between disparity representation and tuning symmetry may reflect how the stereoscopic system has adapted to typical input images experienced in natural viewing conditions. […] Therefore, the even-symmetric neurons with the match-based representation can be viewed as dedicating their disparity encoding to naturally occurring stereoscopic images and thus efficiently coding binocular disparity (Attneave, 1954; Barlow, 1961).”

3) Novelty and significance: Precisely what new has been learned here? This must be spelled out clearly. One gets the impression that the main observations in this study are not new. The authors themselves published work on V4 (some of these data were used in this study), and Krug et al., 2003 described the responses of MT neurons to anticorrelated disparity. (Possibly the only difference being the MT neurons with an area ratio of 1, which were described in Krug et al. but were almost absent in this study.) Perhaps the most novel aspect of this study is the use of traditional ventral pathway stimulus (static RDS and aRDS stimuli) for dorsal pathway study. These results (fast robust response) serve to further support the view of MT/dorsal pathway in action vision. The V4 findings, while well conducted, do not significantly change or extend our view of V4 function in binocular disparity.

This study is about the similarities and differences between V4 and MT, not individual characterizations of each area. Although we used a previously published data set for V4, the study presents three novel findings regarding the MT-V4 relationship. Specifically, our paper:

1) Provides the first convincing evidence for the differential disparity representations between MT and V4. Although this hypothesis has been suspected to be true for some time, we believe that providing direct evidence is scientifically significant.

2) Identifies a novel relationship between disparity representation and tuning symmetry with surprising consistency: within MT, within V4, and across the two areas.

3) Provides the first quantitative comparison of speed/robustness in the disparity signaling between MT and V4 with matched stimuli at the single-unit resolution. An important goal of visual neuroscience is to quantitatively characterize the visual system. In light of this goal, we think that our results are scientifically significant.

Also, from a broader viewpoint, our data set is valuable because it facilitated analyses that:

4) Directly compare single-unit selectivity of dorsal and ventral areas with matched stimuli. Only a few studies have made such a direct comparison, even though it is crucial to improve our understanding of the parallel processing strategy employed by the visual system.

Discussion:

“Differential disparity representations between MT and V4 have been suspected for more than a decade (Parker, 2007), but direct, convincing evidence has been lacking to date. Notably, the available MT and V4 studies with aRDSs used different statistical criteria (Krug et al., 2004; Tanabe et al., 2004). Abdolrahmani et al., 2016, re-analyzed the MT data set in Krug et al., 2004, and compared it against their V4 data set using the same statistical criterion, but failed to find a convincing difference between how the two areas responded to aRDSs (Abdolrahmani et al., Figure 11B). Consistent with this earlier result, we demonstrated that the disparity selectivity for aRDSs was indeed indistinguishable between MT and V4 when examined with the same methods (Figure 5D-F). Only the use of graded anticorrelation and a novel area-ratio metric revealed the MT-V4 difference in the disparity representation.”

2) See our response to the comment 2b.

3) Discussion:

“Also, the disparity selectivity in V4 was, on average, about 30% slower and 30% weaker than that in MT, even though we used the random-dot stimuli tailored to drive V4 neurons.”

4) Conclusion:

“We directly compared single-unit selectivity between dorsal and ventral visual areas using matched stimuli, following the footsteps of a few pioneering studies (Cheng et al., 1994; Lehky and Sereno, 2007).”

4) Overstated conclusions: (a) The conclusion that the data demonstrate “a general principle of the stereoscopic system both within and across the dorsal and ventral pathways” seems too strong based on this study in only two areas. For this conclusion, we need data in many other areas. Nevertheless, I also acknowledge that this will not be easy because most visual areas contain either a strongly match-based representation (such as inferotemporal cortex and parietal areas AIP), or a correlation-based representation such as V1 and V2.

We agree with the reviewer and limited the scope to MT and V4.

“…general principle of the stereoscopic system both within and across the dorsal area MT and ventral area V4”

(b) Similarly, the conclusion that “neurons with even-symmetric tuning preferentially constituted the neural solution to the correspondence problem in extrastriate areas” seems too general.

We revised the sentence to be more specific.

See our response to 2b, novelty issue.

(c) The authors statement that “the correlation-based disparity signals in MT directly contribute to some aspects of depth perception” is too strong since the behavioral effects of microstimulation in MT could have been caused by the match-based neurons alone (since microstimulation does not discriminate between the two types of neurons).

We revised the sentence so the point is more nuanced.

“This implies that the correlation-based disparity signals in MT might directly contribute to some aspects of depth perception, …”

5) Visual eccentricities of recordings in V4 and MT: The mean eccentricities of the V4 neurons are ~9 deg while that of MT neurons are ~7 deg. This seems rather eccentric for V4. Do these findings hold for RFs in more central (0-3 deg) eccentricity? This is important as V4's primary function is in central vision.

We appreciate that the reviewers raised this important point. We do not yet have the direct experimental data of V4’s disparity representation near the fovea. (Foveal stereo experiments are notoriously difficult as stereo stimuli presented at or near the fovea drive vergence eye movements and compromise the measurement of disparity tuning curves. Thus, we think such a challenging experiment warrants a full, independent project.) However, we present two kinds of indirect evidence supporting the hypothesis that neurons with smaller eccentricities have disparity representations that are more strongly match-based. The first type of evidence comes from a novel simulation of the threshold energy model (Doi and Fujita, 2014) with different receptive-field (RF) sizes, while the second comes from a comparison of two previous studies conducted in V1 with different eccentricities. We reported them in the Results section in the revised manuscript (Disparity representation of V4 neurons near fovea, Figure 9, Figure 9—figure supplement 1, Appendix 2—Figure 9—text supplement 1). Also, we revised the text about eccentricity in the Materials and methods section according to the updated

Results.

We cannot agree more with the reviewers’ point that the eccentricity effect on neuronal selectivity is an extremely important topic in visual neuroscience in general. However, we would like to reiterate that the specific scientific question we pursued in the present study concerned the differences between MT and V4. If we were to conduct the V4 experiments near the fovea, the large eccentricity difference between the MT and V4 data sets would have been a confounding variable that would have compromised our ability to properly interpret the data. This leads to the constraint that we could not possibly run our experiments without using less optimal or less conventional stimulus parameters for one or both visual areas.

Results:

“Disparity representation of V4 neurons near the fovea

The ventral pathway plays important roles in foveal vision (Sheth and Young, 2016). However, the mean eccentricity was 9.0° for V4 in our data sets. […] We suggest that this eccentricity dependency on the disparity representation in V1, which was consistent with our simulation results, might be inherited in V4.”

Materials and methods:

“Most of our MT neurons (65 out of 83) had their RFs in the lower visual field; RFs of all of our V4 neurons (N = 78) were located in lower visual field. The mean eccentricity was 6.9° (s.d. = 3.3°) for MT and 9.0° (s.d. = 3.0°) for V4 in our data sets. […] We do not think that the lack of significant correlation convincingly falsified the speculated relationship between eccentricity and disparity representation, because there could be many reasons why the relationship was not detected (e.g., the eccentricity variation was not large enough to detect the relationship; note that in the simulation the size of the largest RF examined was 16 times that of the smallest RF).”